# Medium-term dynamics of a Middle Adriatic barred beach

Matteo Postacchini[1], Luciano Soldini[1], Carlo Lorenzoni[1], and Alessandro Mancinelli[1]

[1]Department of Civil and Building Engineering, and Architecture, Università Politecnica delle Marche, 60131 Ancona, Italy

*Correspondence to:* Matteo Postacchini (m.postacchini@univpm.it)

**Abstract.** In the recent years, attention has been paid to beach protection by means of soft and hard defenses. Along the Italian coasts of the Adriatic Sea, sandy beaches are the most common landscapes and around $70\%$ of the Marche-Region coasts (central Adriatic), is protected by defense structures. The longest free-from-obstacle nearshore area in the region includes the beach of Senigallia, frequently monitored in the last decades and characterized by a multiple bar system, which represents a natural beach defense. The bathymetries surveyed in 2006, 2010, 2011, 2012 and 2013 show long-term stability, confirmed by a good adaptation of an analyzed stretch of the beach to the Dean-type equilibrium profile, though a strong short-/medium-term variability of the wave climate has been observed during the monitored periods. The medium-term dynamics of the beach, which deal with the evolution of submerged bars and are of the order of years or seasons, have been related to the wave climate collected, during the analyzed temporal windows, by a wave buoy located $40\,\mathrm{km}$ off Senigallia. An overall interpretation of the hydrodynamics, sediment characteristics and seabed morphology suggests that the wave climate is fundamental for the morphodynamic changes of the beach in the medium term: time ranges during which waves mainly come from NNE/ESE are characterized by a larger/smaller steepness and by a larger/smaller relative wave height, and seem to induce seaward/shoreward bar migration, as well as bar smoothing/steepening. Moving southeastward, the bar dimension increases, while the equilibrium profile shape suggests the adaptation to a decreasing sediment size in the submerged beach. This is probably due to the presence of both the harbor jetty and river mouth North of the investigated area.

## 1 Introduction

Our communities are experiencing a series of problems and difficulties related to the inundation risk in the coastal areas, the protection of nearshore regions, and the use of beaches for tourist and recreational activities. In the last decades, an increasing attention has been paid to short- and long-term predictions associated with the climate change effects, which will significantly impact on beaches and coastal areas (e.g., see Houghton et al., 2010; Ranasinghe et al., 2013). In fact, such predictions are associated with both the mean sea-level rise and the more frequent sea storms, occurring during both summer and winter. The understanding of the main physical processes driven by such changes is fundamental for (i) the modeling of the nearshore dynamics, including rapid morphological changes to the beach (e.g., Postacchini et al., 2016b), (ii) the correct prediction of coastal flooding (e.g., Villatoro et al., 2014), (iii) the proper design of protection solutions (e.g., Lorenzoni et al., 2016) and (iv) the correct analysis of future scenarios in the coastal area (e.g., see Benetazzo et al., 2012; Lionello et al., 2012).

Several studies (e.g., Benavente et al., 2006; Walton and Dean, 2007) showed that a proper representation of the local bathymetry is fundamental both to correctly predict the seabed changes induced by wave/current forcing and to design efficient solutions for the coastal protection. Hence, typical bedforms of unprotected sandy beaches should be taken into account. In particular, submerged subtidal bars usually form on bottom slopes within 0.005–0.03 and their height ranges between some centimeters to meters (Leont'ev, 2011). In semi-protected and open coasts, two-dimensional longshore bars are quite common and have been extensively studied, though the complex mechanisms of generation and migration are not yet completely understood. Generation of submerged bars can be ascribed to three different mechanisms, i.e. wave breaking, infragravity waves and self arrangement (Wijnberg and Kroon, 2002), while the bar migration depends on several coastal processes and has been investigated in the field (e.g., Ruessink et al., 1998), numerically (e.g., Dubarbier et al., 2015) and through laboratory experiments (e.g., Alsina et al., 2016). It has been observed that swash-zone slope, grain size and wave characteristics play an important role. The influence of the slope on the bar dynamics has only been observed during laboratory experiments, after an *ad hoc* manual reshaping of the swash zone (Baldock et al., 2007; Alsina et al., 2012). On the other hand, field observations confirmed that the grain size could be important in the bar migration rates, due to the larger sediment transport induced by finer sands (Goulart and Calliari, 2013), while the wave characteristics are fundamental for the bar migration direction. In particular, the wave breaking over the bars leads to the generation of a deep return flux, known as undertow, which promotes a seaward motion. As an example, Gallagher et al. (1998) observed, near Duck (North Carolina), an intensified wave breaking occurring over the bar during storms, inducing a large undertow inshore of the bar that pushed it seaward. Conversely, a shoreward bar migration was also observed under small waves, during less energetic states (see also Goulart and Calliari, 2013).

While numerical simulations well reproduced the offshore migration during severe conditions, some difficulties arose when reproducing the onshore bar motion during mild wave conditions (Gallagher et al., 1998; Plant et al., 2004), suggesting that not all the processes involved in the bar migration were clearly understood and correctly simulated, e.g., lower-frequency waves. Further, Ruessink et al. (1998), who analyzed the cross-shore sediment transport and morphological changes occurring in the nearshore area of Terschelling (Netherlands), stated that the role of the infragravity waves has not been completely understood. In particular, it was fairly clear that during energetic conditions, the suspended load dominated over the bedload and the morphodynamics were controlled by undertow and, probably, infragravity waves: such infragravity contribution, more important during breaking than during calm conditions, mobilize large amounts of sediments, which are then advected offshore by the undertow.

The importance of infragravity waves is confirmed by other authors, and a detailed study about their influence on the bar dynamics was undertaken by Aagaard et al. (1994) using field data collected at Stanhope Lane Beach (Canada). They stated that the sediment transport induced by infragravity waves may be either shoreward or seaward, and suspended sediments are mainly transported towards antinodes in the water surface elevation. However, the contribution of infragravity waves on both sediment transport and sandbar motion can be neglected on time scales of years, i.e. when dealing with medium-term morphodynamics (Ruessink and Terwindt, 2000).

With the purpose to characterize the sandbar migration, an important parameter has been recently introduced. This is the local relative wave height, i.e. the ratio between local wave height $H$ and water depth over the bar crest $h_{cr}$. Values smaller than $\sim 0.3$ promote landward migration, while values larger than $0.6$ promote seaward migration (Houser and Greenwood, 2005).

In particular, along the Dutch coast (Ruessink et al., 1998; Ruessink and Terwindt, 2000), a relative wave height $H_s/h_{cr} = 0.33$ represented the onset of breaking, with $H_s$ being the local significant height. Hence, $H_s/h_{cr} > 0.33$ referred to breaking intensification and undertow increase, leading to seaward bar migration. While $H_s/h_{cr} < 0.33$ indicated dominance of short waves and wave skewness, leading to shoreward bar migration. The analysis of the velocity moments and sediment transport confirmed the correlation between medium-term wave conditions and short-term sediment transport measurements (Ruessink and Terwindt, 2000).

From a physical point of view, the increase of both $H_s/h_{cr}$ and breaking intensification produces an increase of the breaking wave celerity (e.g., see Postacchini and Brocchini, 2014), leading to an intensification of the shoreward volume flux, hence to a wave setup (e.g., see Soldini et al., 2009) and to the following increase of the undertow velocity (e.g., see Kuriyama and Nakatsukasa, 2000).

Only few literature studies have been carried out to investigate the seasonal and annual scale of the beach dynamics (e.g., Ruggiero et al., 2009). Some field observations confirmed a cyclic behavior of multiple bars (Ruessink and Terwindt, 2000; Goulart and Calliari, 2013), mainly characterized by three stages, i.e. initial generation, seaward migration and final degradation. Conversely, other authors observed a continuous landward motion, until bar-shore welding, even during storm events (Aagaard et al., 2004). While the offshore migration is promoted by the undertow dominance in the net transport balance, as already stated, the onshore migration is probably enhanced by storm surge: this increases both skewness and phase coupling and reduces the undertow contribution.

The present study describes the seabed evolution of a natural unprotected beach stretch of Senigallia (Marche Region, Italy), a touristic town of the Italian Middle Adriatic. The available bathymetries, covering the last decade, and the wave climate, enable us to analyze the medium-term morphological evolution of the beach, which is of the order of years or seasons and includes the geometry and migration of the submerged bars, as a function of the wave forcing. To the authors' knowledge, this is the first study on the medium-term beach evolution and bar migration occurring in a sandy beach of the Adriatic Sea, a semi-enclosed basin characterized by small tidal excursions ($\sim 40 cm$) and reduced wave heights, if compared to, e.g. the Dutch coastal areas.

The manuscript is divided as follows. Sect. 2 and Sect. 3 illustrate, respectively, the investigated site and the available data. Results are presented in Sect. 4 and discussed in Sect. 5. Some conclusions close the paper.

## 2 Description of the site

The analyzed coast is part of the longest unprotected beach of the Marche Region, which extends from the estuary of the Misa River, whose final reach is highly engineered and adjacent to the Senigallia harbor, to $\sim 3.5\,km$ North of the Esino River estuary, hence for a total length of $\sim 12\,km$ (Fig. 1). As observed during recent field experiments, the Misa River estuary is

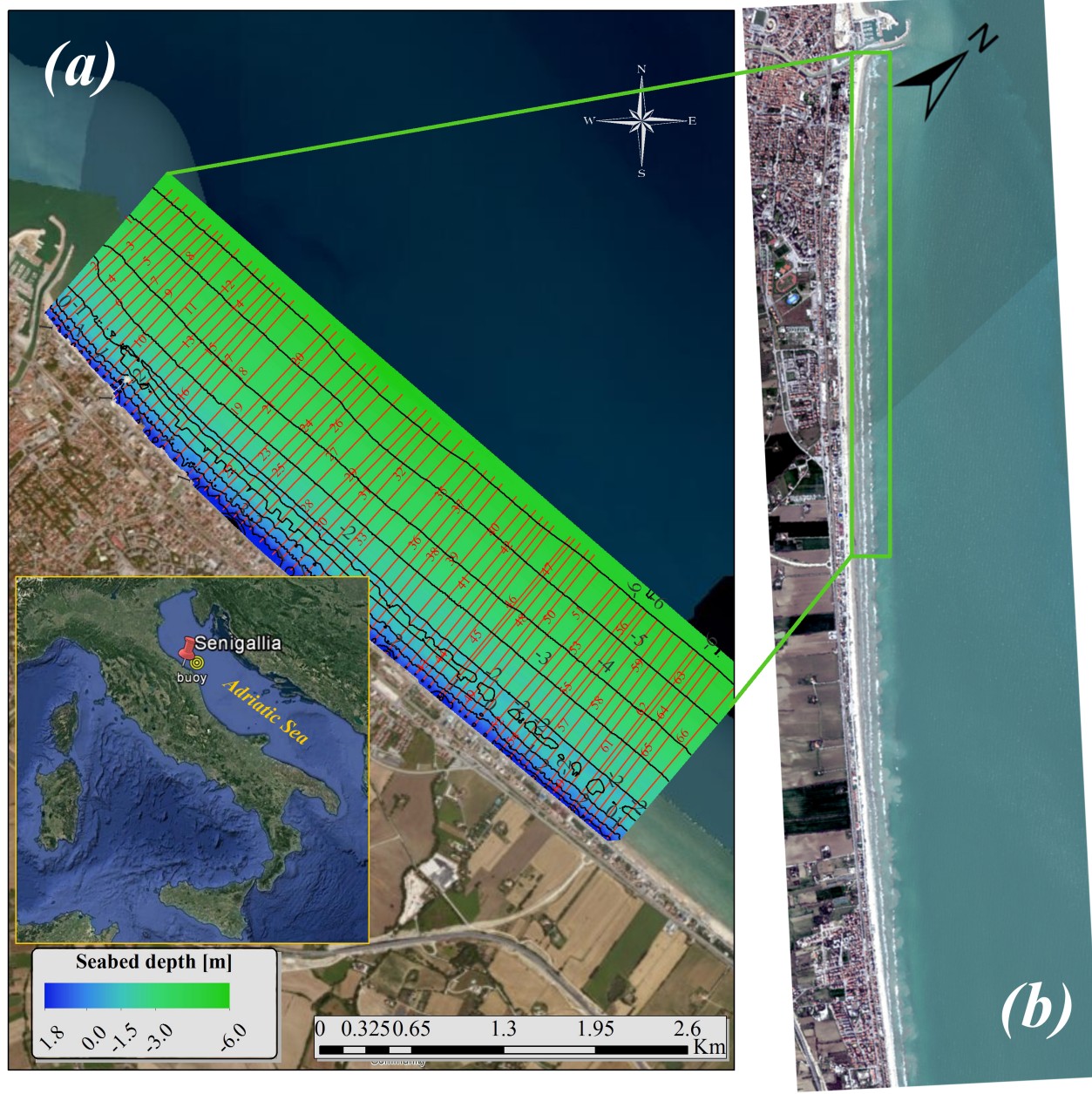

**Figure 1.** Natural beach of Senigallia: **(a)** bathymetry with isobaths and position of cross-shore profiles referring to June 2006 (the white spot between profiles 10 and 11 is the "Rotonda") and **(b)** satellite view of $\sim 10\,\mathrm{km}$ beach South of the harbor/Misa River estuary.

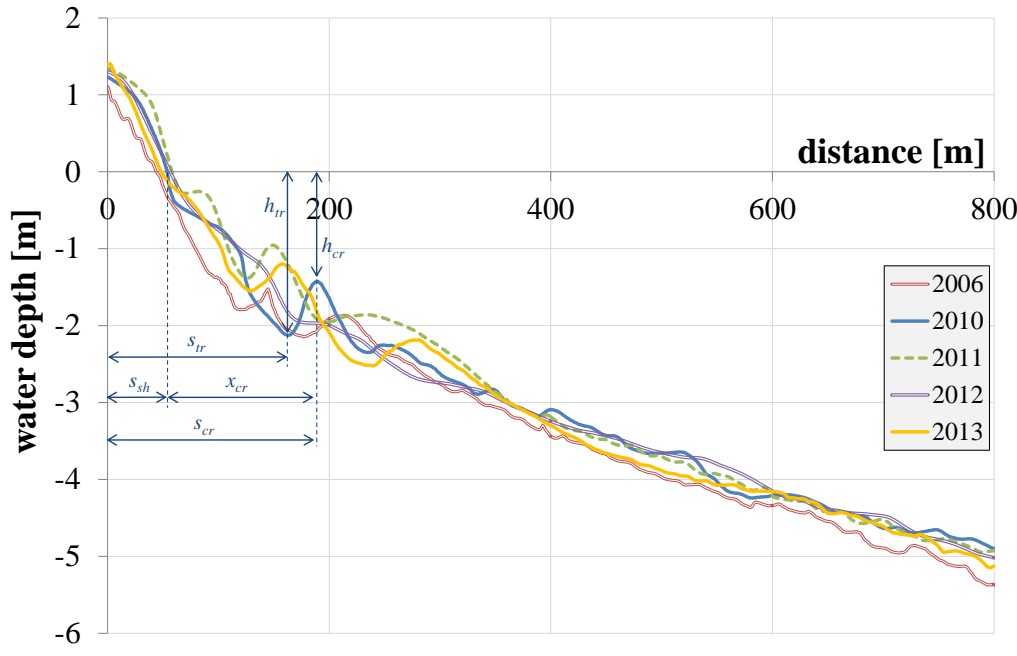

**Figure 2.** Example of cross-shore evolution (profile 13) during the investigated years. Bar characteristics referring to the 2010 profile are also reported.

dynamic throughout the year, especially during sea storms driven by NNE winds, which mobilize a large amount of sediment and generate significant erosion/deposition patterns nearby the rigid structures (Brocchini et al., 2015, 2017). The investigated site is characterized by a swash-zone slope in the range $1:30$–$1:40$, an array of submerged bars in a water depth $h = 0$–$3\,\mathrm{m}$, and a mild slope of about $1:200$ for $h > 3\,\mathrm{m}$ (Fig. 2). The emerged beach is characterized by fine ($d_{50} = 0.125$–$0.25\,\mathrm{mm}$) and

5    medium ($d_{50} = 0.25$–$0.5\,\mathrm{mm}$) sands, with fine sand in the submerged part.

The wave climate in the investigated area was obtained from a waverider of the Italian wave measurement network (RON), located $\sim 23\,\mathrm{nm}$ East-North-East of Senigallia. It worked between March 1999 and March 2006 and between December 2009 and November 2013, the data between 2006 and 2010 surveys thus missing. During the 11 years recordings, the waves mainly came from ESE, NNE and NW (Fig. 3a), the main events being induced by Bora (coming from NNE) and Levante-Scirocco

10   (from ESE) winds. The wave frequency (blue outline) is better distributed throughout the directions, while the wave energy (orange area) is characterized by sharper peaks corresponding to ESE and NNE.

The analysis of the beach morphology, using the concept of the equilibrium beach profile (Dean, 1991), describes the long-term beach equilibrium of a natural beach, i.e. the balance between erosive and accretive forcing, through

$$h = Ax^{2/3}, \tag{1}$$

where $h$ is the water depth and $x$ the distance to shoreline. $A$ is a dimensional shape parameter, directly related to the median grain diameter $d_{50}$ (Hanson and Kraus, 1989). Notice that Eq. 1 also leads to the estimate of the so-called "fitting depth", i.e. the water depth at which the measured profile collapse over the equilibrium profile. Though recent models account for further parameters, like seasonal changes (Inman et al., 1993) or the generation of submerged bars (Holman et al., 2014), their application is fairly difficult and it has been demonstrated that Eq. 1 properly represents the long-term natural profile, to be used for coastal engineering purposes (e.g., Walton and Dean, 2007; Soldini et al., 2013). With the purpose to estimate a proper fitting depth, the submerged beach, surveyed in 2006, 2010, 2011, 2012 and 2013 up to a depth of $\sim 6m$ (see also Sect.3), has been extended up to $10m$ assuming as constant the mild slope characterizing the deeper beach stretch, i.e. $1:200$. Using either the least-square approach or the continuity of volume, i.e. integration of Eq. (1), the results are similar. From the Digital Terrain Model (DTM) of Fig. 1a, referring to the 2006 survey, 66 profiles have been extracted. It is important to notice that $A$, and similarly $d_{50}$, decreases moving southward. The largest values occur close to the Senigallia harbor (profile 1 of Fig. 1a), where $A \sim 0.069$ and, following Hanson and Kraus (1989), $d_{50} \sim 0.15\,\mathrm{mm}$, while the smallest occur $\sim 3.9\,\mathrm{km}$ South of the harbor (profile 66), where $A \sim 0.060$ and $d_{50} \sim 0.13\,\mathrm{mm}$. Such values are in agreement with the fine sand characterizing the submerged beach (Lorenzoni et al., 1998a). It has been observed that, throughout the coast surveyed in 2006, the natural beach well adapts to the Dean-type equilibrium profile. This is confirmed by the following campaigns (2010–2013), when a good adaptation still exists, the values of $A$ remain almost constant in time and decrease moving southward. Further, the fitting depth increases from the harbor to the "Rotonda", i.e. the pile-mounted permeable structure within profiles 10 and 11, and decreases South of the "Rotonda". This suggests a sediment motion occurring at larger depths in correspondence of the structure, that partially (and locally) influences the beach evolution and bar migration.

Although the present study aims at investigating the nearshore area, where both cross-shore and alongshore sediment transport contributions determine the short- to long-term equilibrium of the shallow beach, a regional framework may also be taken into account. In general, the sediment transport throughout the Adriatic Sea is influenced by a number of factors. Specifically, the western Adriatic coast is characterized by large depositions nearby the rivers (e.g., at the Misa River estuary, as described by Brocchini et al., 2017) and especially close to the Po Delta. Further depositions occur north of the Gargano Peninsula, due to the Western Adriatic Coastal Current (WACC, e.g., see Harris et al., 1998; Sherwood et al, 2004), which is responsible of the suspended sediment transport. In the same regional framework and for depths greater than $10m$, while Bora-induced waves provide large sediment fluxes, Scirocco-induced waves lead to sediment flux reduction, though sediment suspension increases due to significantly energetic conditions.

## 3 Experimental data

The natural beach of Senigallia was characterized by a number of bathymetric surveys since the 80s. More recently, due to a specific requirement of the Marche Region, a detailed survey of the nearshore region of Senigallia was undertaken in June 2006, both North and South of the harbor, such areas being respectively characterized by a protected and an unprotected beach. The surveys cover the nearshore region up to a depth of $6\,\mathrm{m}$ and a total length of $4.3\,\mathrm{km}$, most of which ($\sim 3.9\,\mathrm{km}$) South of the harbor (Fig. 1a).

Between 2010 and 2013, after the modification of the harbor entrance, annual bathymetric surveys up to a depth of $6\,\mathrm{m}$ were carried out by the municipality of Senigallia on a $2.5\,\mathrm{km}$-long area covering part of the protected and part of the unprotected beaches.

The available bathymetric surveys enabled us to extract 18 cross-shore profiles which characterize the unprotected beach for about $1\,\mathrm{km}$. The bathymetries have been used for the analysis of the wave-climate-induced morphological changes, i.e. bed variations between two consecutive surveys, in terms of bar migration and geometry. It is worth noting that bathymetries could have been surveyed just after an intense storm, which promotes significant morphological changes. However, the medium-term climate is the sum of a number of energetic and calm states occurring between two consecutive surveys. Hence, a bathymetry does not depend on any specific event, nor on a specific season/month (e.g., see the significantly different beach profiles surveyed in February 2010 and February 2011, illustrated in Fig. 2), but on the sum of the contributions of all such events to the overall morphological change observed in the chosen time range. It is worth noting that the observed daily migration rate, i.e. that related to short-term events, of submerged bars in sandy beaches varies in the range $(1-50)\,\mathrm{m/day}$, while the yearly rate, i.e. that related to medium-term events, is in the range $(0.1-0.35)\,\mathrm{m/day}$ (van Enckevort and Ruessink, 2003). Such values depend on the storm characteristics, but also on the basin type, hence migration rates in the ocean are generally different from those observed in the sea (see also Parlagreco et al., 2011). Further, the separation between the morphological effects induced by long-term and short-term events is difficult, especially in semi-enclosed basins like the Adriatic Sea, which is characterized by an extremely variable climate, with significantly large deviations of the wave characteristics from the mean values, even during a storm. Hence, the aim of the present work is that of analyzing the morphological changes and discussing the cumulative effect of all events occurring between consecutive surveys. Such an analysis is also useful to demonstrate that the beach evolution can be predicted when a limited number of surveys is available, a typical condition for coastal municipalities.

From the analysis of both surveys and satellite data, the submerged bars remain for a stretch of $\sim 12\,\mathrm{km}$. Further, moving southeastward, the sediment size changes, with a transition from sand to gravel occurring $\sim 6\,\mathrm{km}$ South of the harbor (Lorenzoni et al., 1998b). Hence, the initially two-dimensional longshore bars of the investigated area get closer to the shoreline, thus switching to three-dimensional (see Fig. 1b, where the location of the bars is highlighted by both foam and suspended sediment induced by the waves breaking over them). However, the $\sim 1\,\mathrm{km}$-long area South of the harbor can be taken as representative of the sandy beaches characterizing the Middle Adriatic Sea and will be analyzed in the next sections.

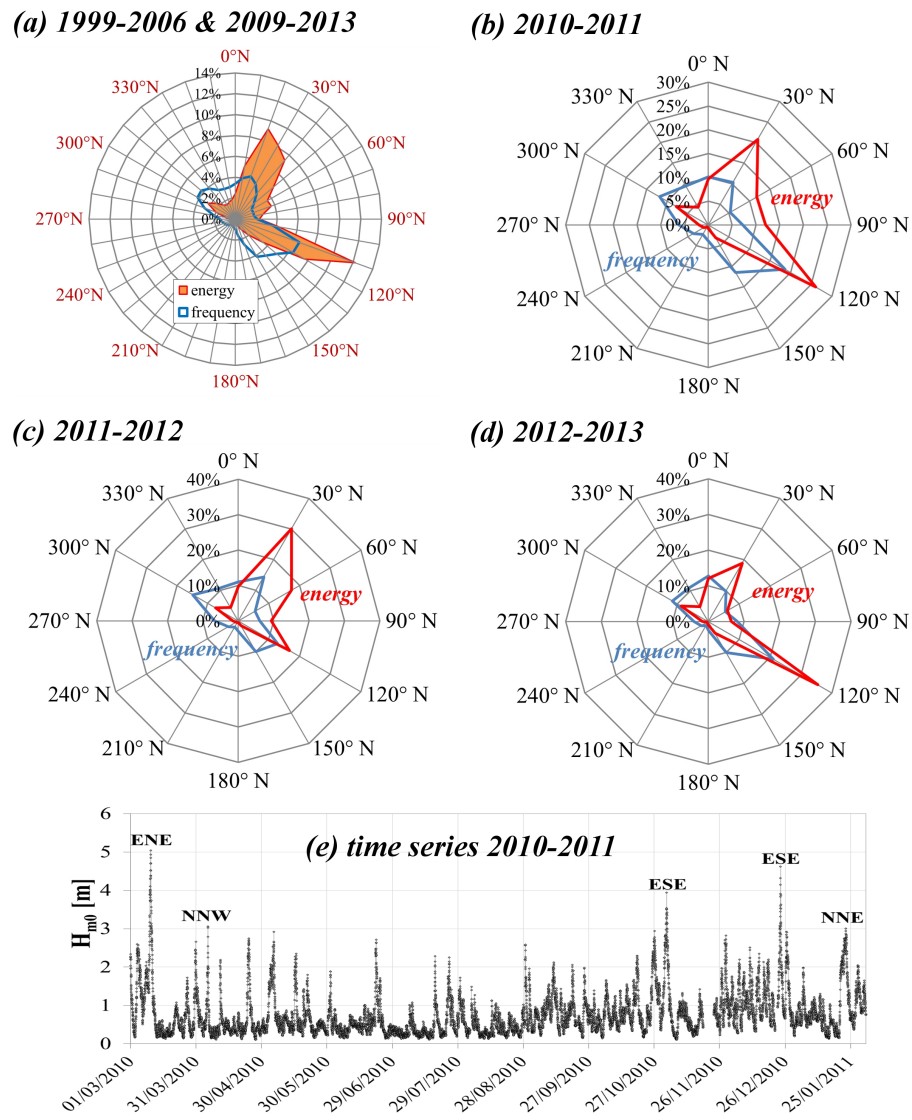

**Figure 3.** Wind roses of wave energy (red line) and frequency (blue line) referring to time periods: **(a)** 1999–2006 and 2009–2013, **(b)** 2010–2011, **(c)** 2011–2012 and **(d)** 2012–2013. **(e)** Time series of the significant height measured between March 2010 and January 2011, with indications of the largest wave ($H_{m0} > 3m$) direction.

## 4   Results

The following sections illustrate the results obtained from the analysis of the seabed variation using the available bathymetric surveys, which refer to June 2006, February 2010, February 2011, April 2012 and May 2013, and the related wave climate. Both migration and geometry of the submerged bars are discussed.

### 4.1   Wave climate

Except for the period 2006–2010, during which the waverider did not work, the wave climate referring to the considered time periods is illustrated in Fig. 3: both the overall climate (Fig. 3a) and the single-period climates (2010–2011 in Fig. 3b; 2011–2012 in Fig. 3c; 2012–2013 in Fig. 3d) are shown. Notice that the wave climate has been evaluated using the spectral data (e.g., significant height, peak period) provided by the waverider every half an hour. The most frequent and the most energetic waves are, in both cases, those coming from either ESE, i.e. forced by Levante-Scirocco winds, or NNE, i.e. forced by Bora winds, which thus correspond to the predominant waves of such a coastal area. Waves from NW are frequent as well, but less energetic. While the wave frequency (blue lines) is fairly well distributed and homogeneous, the wave energy (red lines) is characterized by sharper peaks in correspondence of the dominant directions and by a reduced distribution elsewhere.

It is well known that the wave climate for the extra-tropical regions at intermediate latitudes, like that of the Adriatic Sea, is characterized by the presence, at the soil level, of closed dynamical systems, as cyclones and anticyclones. Usually, soil weather systems are connected to a movement with an upper-level wavy structure, that slowly migrates eastward. So, the presence of migrating temporal troughs and ridges alternates during the year. Troughs are linked to low atmospheric pressure areas, with colder air and a sequence, usually, of cyclones. Ridges are linked to high pressure areas, with warmer air and anticyclonic, more stable, weather. Specifically, the Bora is a cold and dry wind usually linked to a well-developed anticyclone on the central or northern Europe and a relative low pressure on the Mediterranean Sea. It is more frequent and very intense during the winter. Conversely, the Scirocco is a southern warm wind, which is dry in Africa, then becomes wet passing on the Mediterranean Sea, and finally generates big sea storms with important surges and persistent swell. Scirocco intensities are less than the Bora, but generate longer and more enduring waves.

In the studied site, the weather is not characterized by two distinct (seasonal) behaviors, rather by a pronounced temporal variability of the wave climate during the year: the two peaks illustrated in Fig. 3a-d do not refer to the prevalent conditions occurring, respectively, in summer and winter, but mainly refer to the most severe winter storms (Fig. 3e), the summertime being characterized by milder wave conditions, due to less strong winds and slowly changing wind directions during storms (see also Brocchini et al., 2015). Further, the fairly well distributed frequency, with respect to the more peaked energy flux, indicates that the annual variability of storms is not bound to the seasonal variability of wave climate. This can be observed in Fig. 3e, where the time series of the significant wave height recorded by the waverider in 2010-2011 is illustrated. The incoming direction of the storms characterized by $H_{m0} > 3m$ is also reported. Notice that three out of five large storms occurred in winter, coming respectively from ENE (10/03/2010), ESE (23/12/2010) and NNE (22/01/2011). The alternation in the winter-storm

direction is confirmed by Brocchini et al. (2017), who observed two consecutive storms in January 2014, the first due to Bora winds and the second, after three days, due to winds coming from WNW and N.

As a consequence of the elongated shape of the Adriatic Sea (see Fig. 1 and Sect. 3), significantly different physics generate in the studied area during NNE and ESE storms. Starting from this assumption, our methodology elaborates all available buoy data with the purpose to get a wave-climate descriptor which accounts for the main processes occurring during the considered time intervals. In particular, with reference to both frequency and energy flux, a statistic analysis of the main sectors has been undertaken for each selected time period, as in the following steps:

- the wave climate during the whole time range is analyzed to obtain the energy distribution illustrated in Fig. 3b-d,

- the most energetic direction is chosen and associated to a specific sector, i.e. $(105–135)°$ for ESE or $(15–45)°$ for NNE,

- the waves falling in the chosen sector are analyzed to get the most energetic wave-height ranges,

- the most frequent wave-period ranges associated to such heights are chosen.

In detail, since Fig. 3b and Fig. 3d show that the ESE forcing dominates in 2010–2011 and 2012–2013, only the $(105–135)°$ sector has been analyzed. Conversely, the NNE forcing dominates in 2011–2012, hence this has been associated to $(15–45)°$. In the ESE cases, the largest energetic contributions (more than $60\%$ of the total) are ascribed to significant wave heights in the range $H_{m0} = (1–3)\,\text{m}$ (2010–2011) and $H_{m0} = (1.5–3.5)\,\text{m}$ (2012–2013). The most frequent waves falling in such ranges are characterized by mean periods $T_m = (4–5.5)\,\text{s}$ (2010–2011) and $T_m = (4.5–6)\,\text{s}$ (2012–2013). Peak periods are, respectively, $T_p = (6–7.5)\,\text{s}$ and $T_p = (7–8.5)\,\text{s}$. In 2011–2012, the largest energetic contribution ($> 60\%$) belongs to a narrower wave-height range, i.e. $H_{m0} = (1–2.5)\,\text{m}$, which corresponds to most frequent waves falling in wider ranges $T_m = (3.5–5.5)\,\text{s}$ and $T_p = (5–7)\,\text{s}$.

With the purpose of characterizing each time interval with specific wave features, the most energetic direction (ESE or NNE) associated with the most probable wave-height class gives the most probable wave-period class. As an example, Tab. 1 shows that in 2012-2013 the largest energy-flux distributions characterize the ranges $H_{m0} = 1.5–2m$ ($16.56\%$) and $H_{m0} = 3–3.5m$ ($16.02\%$). However, we believe that the range $H_{m0} = 1.5–2m$ is more representative, as more probable height-period classes exist (see Tab. 2). In particular, $10.51\%$ of all waves are characterized by $H_{m0} = 1.5–2m$ and $T_m = 5.0–5.5s$, while waves with $H_{m0} = 3–3.5m$ are not so frequent.

**Table 1.** Energy-flux distribution (%) in 2012–2013 (only referring to sector 105–135°). The most probable class is reported in bold.

| | | | | | $H_{m0}$ [m] | | | | | |
|---|---|---|---|---|---|---|---|---|---|---|
| 0.0–0.5 | 0.5–1.0 | 1.0–1.5 | **1.5–2.0** | 2.0–2.5 | 2.5–3.0 | 3.0–3.5 | 3.5–4.0 | 4.0–4.5 | 4.5–5.0 | >5.0 |
| 0.00 | 0.00 | 10.41 | **16.56** | 12.37 | 15.03 | 16.02 | 10.87 | 5.43 | 9.85 | 3.45 |

The described procedure leads to the following values, which represent the most probable combinations $(H_{m0}, T_m)$ and $(H_{m0}, T_p)$, related to the most energetic waves (ESE or NNE).

**Table 2.** Frequency (%) for classes of $H_{m0}$ and $T_m$ in 2012–2013 (only referring to sector 105–135°). The most probable classes are reported in bold.

| $T_m$ [s] | $H_{m0}$ [m] | | | | | | | | | | |
|---|---|---|---|---|---|---|---|---|---|---|---|
| | 0.0–0.5 | 0.5–1.0 | 1.0–1.5 | **1.5–2.0** | 2.0–2.5 | 2.5–3.0 | 3.0–3.5 | 3.5–4.0 | 4.0–4.5 | 4.5–5.0 | >5.0 |
| <2.00 | 0.00 | 0.00 | 0.00 | 0.00 | 0.00 | 0.00 | 0.00 | 0.00 | 0.00 | 0.00 | 0.00 |
| 2.0–2.5 | 0.00 | 0.00 | 0.00 | 0.00 | 0.00 | 0.00 | 0.00 | 0.00 | 0.00 | 0.00 | 0.00 |
| 2.5–3.0 | 0.00 | 0.00 | 0.00 | 0.00 | 0.00 | 0.00 | 0.00 | 0.00 | 0.00 | 0.00 | 0.00 |
| 3.0–3.5 | 0.00 | 0.00 | 0.56 | 0.00 | 0.00 | 0.00 | 0.00 | 0.00 | 0.00 | 0.00 | 0.00 |
| 3.5–4.0 | 0.00 | 0.00 | 7.61 | 0.00 | 0.00 | 0.00 | 0.00 | 0.00 | 0.00 | 0.00 | 0.00 |
| 4.0–4.5 | 0.00 | 0.00 | 11.86 | 1.90 | 0.00 | 0.00 | 0.00 | 0.00 | 0.00 | 0.00 | 0.00 |
| 4.5–5.0 | 0.00 | 0.00 | 9.40 | 7.72 | 0.56 | 0.00 | 0.00 | 0.00 | 0.00 | 0.00 | 0.00 |
| **5.0–5.5** | 0.00 | 0.00 | 8.39 | **10.51** | 3.47 | 0.78 | 0.00 | 0.00 | 0.00 | 0.00 | 0.00 |
| 5.5–6.0 | 0.00 | 0.00 | 2.13 | 6.82 | 4.59 | 2.80 | 0.45 | 0.00 | 0.00 | 0.00 | 0.00 |
| 6.0–6.5 | 0.00 | 0.00 | 0.00 | 1.01 | 2.80 | 3.02 | 2.68 | 1.12 | 0.00 | 0.00 | 0.00 |
| 6.5–7.0 | 0.00 | 0.00 | 0.00 | 0.00 | 0.00 | 1.79 | 2.01 | 0.78 | 0.45 | 0.11 | 0.00 |
| 7.0–7.5 | 0.00 | 0.00 | 0.00 | 0.00 | 0.00 | 0.00 | 1.01 | 1.12 | 0.22 | 0.89 | 0.22 |
| 7.5–8.0 | 0.00 | 0.00 | 0.00 | 0.00 | 0.00 | 0.00 | 0.00 | 0.00 | 0.45 | 0.56 | 0.22 |
| 8.0–8.5 | 0.00 | 0.00 | 0.00 | 0.00 | 0.00 | 0.00 | 0.00 | 0.00 | 0.00 | 0.00 | 0.00 |
| 8.5–9.0 | 0.00 | 0.00 | 0.00 | 0.00 | 0.00 | 0.00 | 0.00 | 0.00 | 0.00 | 0.00 | 0.00 |
| 9.0–9.5 | 0.00 | 0.00 | 0.00 | 0.00 | 0.00 | 0.00 | 0.00 | 0.00 | 0.00 | 0.00 | 0.00 |
| 9.5–10.0 | 0.00 | 0.00 | 0.00 | 0.00 | 0.00 | 0.00 | 0.00 | 0.00 | 0.00 | 0.00 | 0.00 |
| >10.0 | 0.00 | 0.00 | 0.00 | 0.00 | 0.00 | 0.00 | 0.00 | 0.00 | 0.00 | 0.00 | 0.00 |

- 2010–2011: ESE, $H_{m0} = 1.75\,\mathrm{m}$, $T_m = 5.25\,\mathrm{s}$, $T_p = 7.25\,\mathrm{s}$;

- 2011–2012: NNE, $H_{m0} = 2.25\,\mathrm{m}$, $T_m = 5.25\,\mathrm{s}$, $T_p = 6.75\,\mathrm{s}$;

- 2012–2013: ESE, $H_{m0} = 1.75\,\mathrm{m}$, $T_m = 5.25\,\mathrm{s}$, $T_p = 7.25\,\mathrm{s}$.

As expected, due to the available fetch length (see also Fig. 1a), a larger wave steepness ($H_{m0}/L_{p0}$, where $L_{p0}$ is the deep-water peak wavelength) occurs during the NNE-dominated periods.

It is worth noting that in 2010-2011, though ESE is the most energetic direction ($\sim 26\%$), the NNE contribution ($\sim 21\%$) is also important (Fig. 3b). The analysis of the NNE direction suggests that the most energetic waves are characterized by a reduced height range, i.e. $H_{m0} = (1.5\text{–}2.5)\,\mathrm{m}$, associated to periods $T_m = 3.5\text{–}5.5s$ and $T_m = 5.5\text{–}7s$. If we look at the values of the most energetic and frequent classes, we get

- 2010–2011: NNE, $H_{m0} = 2.25\,\mathrm{m}$, $T_m = 5.25\,\mathrm{s}$, $T_p = 6.75\,\mathrm{s}$.

Such a result demonstrates that the NNE sector provides waves steeper if compared to the ESE sector, whether or not it represents the most energetic sector.

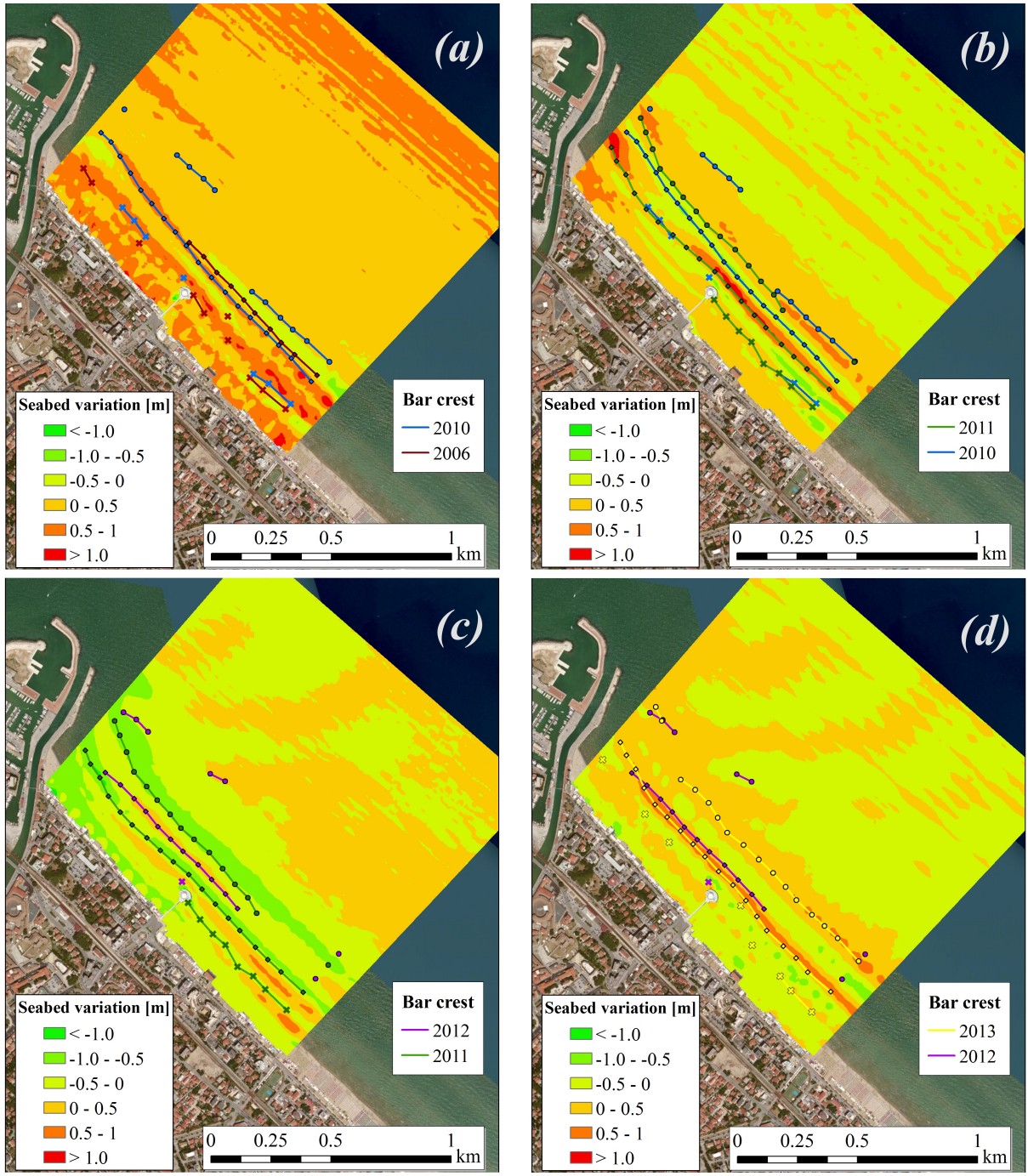

**Figure 4.** Sea bottom variation within time periods: **(a)** 2006–2010, **(b)** 2010–2011, **(c)** 2011–2012 and **(d)** 2012–2013. The bar-crest locations extracted from the cross-shore profiles are represented using colored lines and symbols (+: inner, □: intermediate, ○: outer).

## 4.2 Bathymetric surveys

The available bathymetries have been overlapped using ArcGIS software and the difference in the bed depth has been estimated between each pair of consecutive surveys. Hence, Fig. 4 illustrates the difference between the bed depth measured in 2010 and that measured in 2006 (a), 2011 and 2010 (b), 2012 and 2011 (c), 2013 and 2012 (d). Each case shows seabed patterns which are mostly parallel to the coast. Such parallel patterns illustrate the different location of the submerged bars and their migration through each time interval. In each panel, positive/negative values mean that a seabed accretion/erosion occurred during the considered time period. Large positive values (red patterns) indicate either the filling of the bar trough or the location of the bar crest at the end of the time period (e.g., see the longshore distribution of positive values in Fig. 4a, b, and d, these representing the crest location in 2010, 2011, 2013, respectively). Further, large negative values (green patterns) may also indicate a bar-crest smoothing and a general beach flattening, as shown in Fig. 4c. Notice that the largest variations occur in the nearshore area, i.e. for bed depths smaller than $3m$.

The shoreline is fairly stable and, in the medium-term, oscillates in the cross-shore direction less than $20\,\mathrm{m}$ (Fig. 5a), with the largest motions occurring in 2006–2010 (advance) and 2011–2012 (retreat). To properly reconstruct the bar migration, the crest locations are overlapped to the color maps of Fig. 4.

Further, each of the 18 cross-shore profiles have been characterized by means of (Fig. 2): (i) the shoreline position from a fixed point ($s_{\mathrm{sh}}$), (ii) the distance of each bar crest from both fixed point ($s_{\mathrm{cr}}$) and shoreline ($x_{\mathrm{cr}} = s_{\mathrm{cr}} - s_{\mathrm{sh}}$), and (iii) the bar geometry, i.e. crest ($h_{\mathrm{cr}}$) and trough ($h_{\mathrm{tr}}$) depths. The location of both bar crest $s_{\mathrm{cr}}$ and shoreline $s_{\mathrm{sh}}$ are illustrated in Fig. 5a. Since it is evident (Fig. 4) that a well-defined inner bar only characterizes the 2011 survey, in Fig. 5a we prefer to only analyze the migration of intermediate ($\square$) and outer ($\circ$) bars. In detail, the intermediate bar seems to move slightly shoreward between 2006 (red lines) and 2010 (blue lines), while the outer bar is not evident in 2006. While in 2006 the bar develops between profiles 9 and 18 (see also the light green pattern in the map of Fig. 4a, indicating a bed erosion occurring between 2006 and 2010), in 2010 the bar develops throughout the analyzed domain, being highlighted by the reddish pattern in the map, indicating a bed accretion. In the following period (2010-2011), the shoreward migration of the long intermediate bar (blue lines for 2010, green lines for 2011) is confirmed by the yellow and reddish patterns underneath the 2011 bar-crest alignment (Fig. 4b). Large bar accretions (i.e. variations $> 1m$), which suggest a local bar steepening, do occur nearby the structures. Discontinuities may be observed on the outer bars of 2010, 2012 and 2013 (Fig. 5a). In 2012, both inner and outer bars are partially destroyed. The seaward migration of the intermediate bar (green lines for 2011, purple lines for 2012, Fig. 5a) is highlighted by a bed variation $< -1m$ under the 2011 bar (green pattern in the map of Fig. 4c) and slight accretions ($0 - 0.5m$) under the 2012 bar (orange pattern). Larger accretions located South of the "Rotonda" (reddish patterns) mean that the 2011 bar troughs have been filled in the 2011-2012 period, the resulting beach being flatter than that observed from the other surveys. The final time range shows that the inner and outer bars regenerate in 2013 (yellow lines, Fig. 4d). The intermediate bar seems to move shoreward, this mainly meaning a positive bed variation (reddish pattern) observed in the map under the 2013 bar. Hence, a shoreward migration of the bars occurred in 2006–2010, 2010–2011 and 2012–2013, while a seaward motion only occurred between 2011

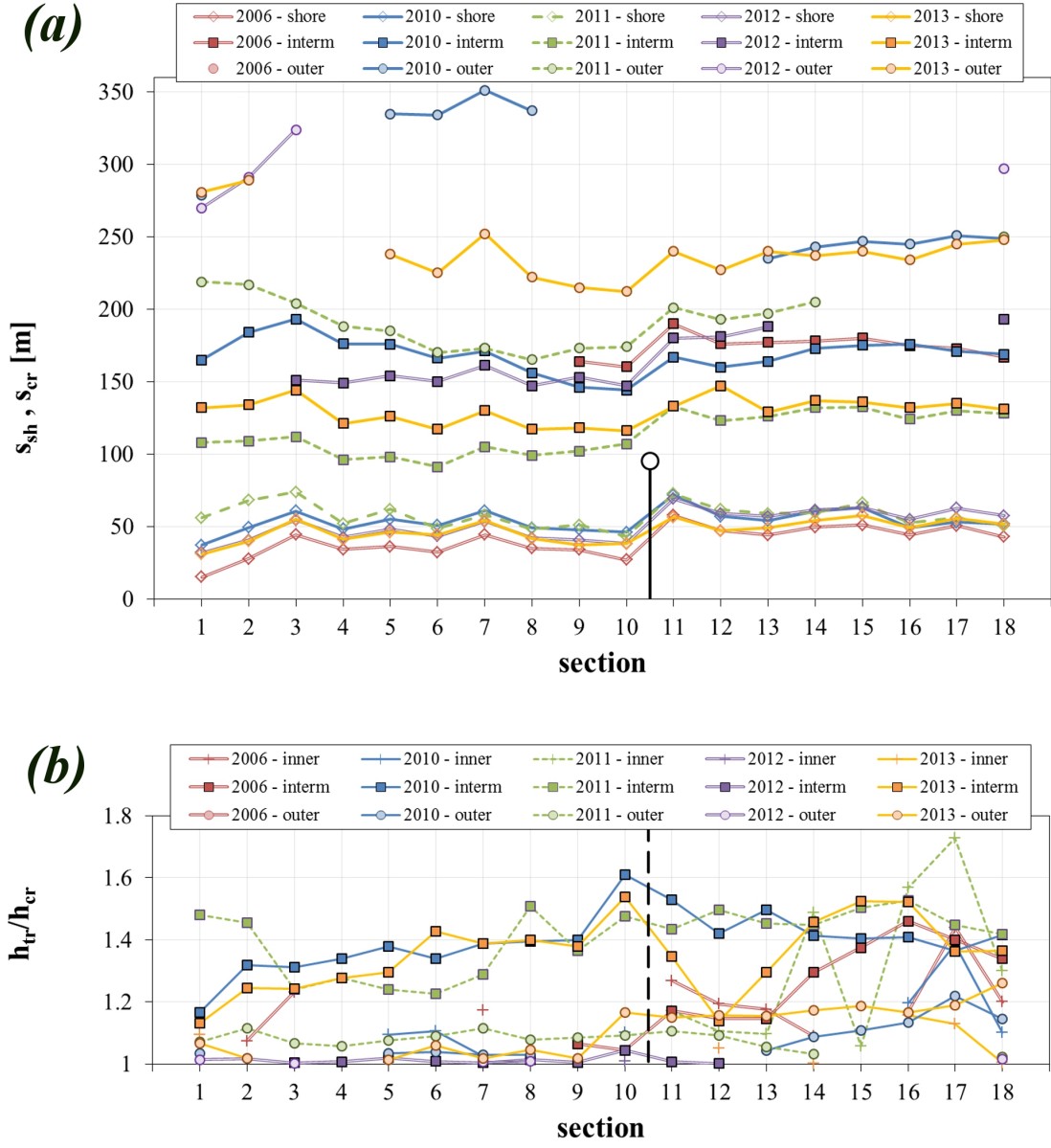

**Figure 5.** Longshore evolution of bar features: (a) shoreline, middle and offshore bars, (b) ratio between trough and crest depth. The vertical black lines represent the "Rotonda" location.

and 2012, when the bars were partially destroyed. After 2012, a partial bar regeneration occurred. In addition, in 2010 the outer bar exists only South of the "Rotonda", in 2011 only North.

Close to the structure, the ratio between trough and crest depths $h_{tr}/h_{cr}$ (Fig. 5b) oscillates within the range $1 - 1.8$. The middle bars ($\square$) show almost regular, slightly varying, trends between profiles 3 and 9, i.e. where they are sufficiently far from both structures, with the bar trough being $25 - 40\%$ deeper than the crest. This occurs for all years, except for 2012, when crest and trough depths were very similar ($h_{tr}/h_{cr} \sim 1$) as the bar was almost completely destroyed. South of the permeable structure, $h_{tr}/h_{cr}$ varies in different ways during the analyzed periods, i.e. it rapidly grows in 2006 and 2013, or remains almost constant in 2010 and 2011. However, it tends to stabilize around 1.4–1.5 at profiles 17–18. Further, the outer bars ($\circ$) are affected by small local changes between profiles 10 and 12 (Fig. 5a), while the depth ratio slightly increases moving South (Fig. 5b). It is worth noting that Fig. 5, i.e. both bar alignment and geometry, invokes the existence of two regions where bars behave differently, one North (between profiles 4 and 9) and one South (profiles 14-18) of the "Rotonda". This also means that the complex hydrodynamics and beach morphology around the structures lead to discontinuities, like those observed for the outer bar in 2010 and 2012, and different beach responses in the two regions (e.g., see Shand et al, 2001). Transition regions also exist, one close to the jetty (profiles 1-4), the other close to the "Rotonda" (profiles 10-13).

While the depth variation of Fig. 4 is representative of the volume changes occurred at each point of the domain, the cross-shore profiles at different alongshore locations more clearly illustrate the volume changes occurred between two consecutive surveys. In particular, since the ESE forcing slightly dominates in 2010-2011 and Fig. 3b suggests a bimodal behavior of the wave climate, three profiles collected in 2010 (blue line) and 2011 (green line) are analyzed. They represent the region located between the jetty and the "Rotonda" (Fig. 6a, profile 6), that around the "Rotonda" (Fig. 6b, profile 10) and that far from the "Rotonda" (Fig. 6c, profile 18). In addition, the cumulative volume change is illustrated (red dashed line), with the aim to explain how the sediment is transported through the cross-shore profile, but notice that the alongshore sediment losses are not accounted for in such an approach. The volume change ($VC$) at the $j$-th cross-shore location ($j = 1...n$, with $n$ the number of points along the $x$ axis), referring to two profiles surveyed at times $k_f$ and $k_i$, may be expressed as

$$VC_j^{(k_f - k_i)} = VC_{j-1}^{(k_f - k_i)} + \Delta V_j^{(k_f - k_i)}, \tag{2}$$

where the volume variation at the $j$-th location between $k_i$ and $k_f$ is

$$\Delta V_j^{(k_f - k_i)} = V_j^{k_f} - V_j^{k_i}, \tag{3}$$

with the volume (per unit length) at the $j$-th location $V_j$ being calculated as the product between the profile discretization in the cross-shore direction ($\Delta x$) and the profile elevation with respect to a horizontal reference system ($z_{b,j}$): $V_j = z_{b,j}\Delta x$. In the example of Fig. 6, time indexes are $k_f = 2011$ and $k_i = 2010$. Notice that at $j = 1$, i.e. $x = 0$, the volume change is $VC_1^{(k_f - k_i)} = 0$.

Along the undisturbed profile (Fig. 6c), the volume change is positive between $x = 0$ and $x \sim 120m$, i.e. up to the 2010 inner bar, suggesting an increase of the upper beach and nearshore area due to that bar and an overall sediment balance in the range $x = 0 - 120m$, as $VC|_{x \sim 120m} = 0$. Further, the volume change is positive also between $x \sim 150$ and $x \sim 210m$, i.e. up to the 2010 intermediate bar, and between $x \sim 210$ and $x = 800m$. Such an analysis suggests that three distinct regions exist. The first region is characterized by both the migration of the inner bar and the increase of the upper beach and nearshore area.

The migration of the intermediate bar occurs in the second region. The third region deals with a significant beach reshaping involving the outer bar. Similar results have been found for the profile surveyed between the rigid structures (Fig. 6a), where both inner and intermediate bars contribute to the nearshore/upper beach change. Finally, the sediment balance throughout the analyzed profile significantly changes close to the "Rotonda" (Fig. 6b), as the volume change never goes to zero. In particular, the upper beach change mainly depends on the inner bar, as the volume accretion (i.e. $VC$ increase) occurring at $x = 0 - 90m$ partially derives ($\sim 39\%$) from the volume erosion (i.e. $VC$ decrease) occurring at $x = 90 - 115m$. Hence, close to the structure,

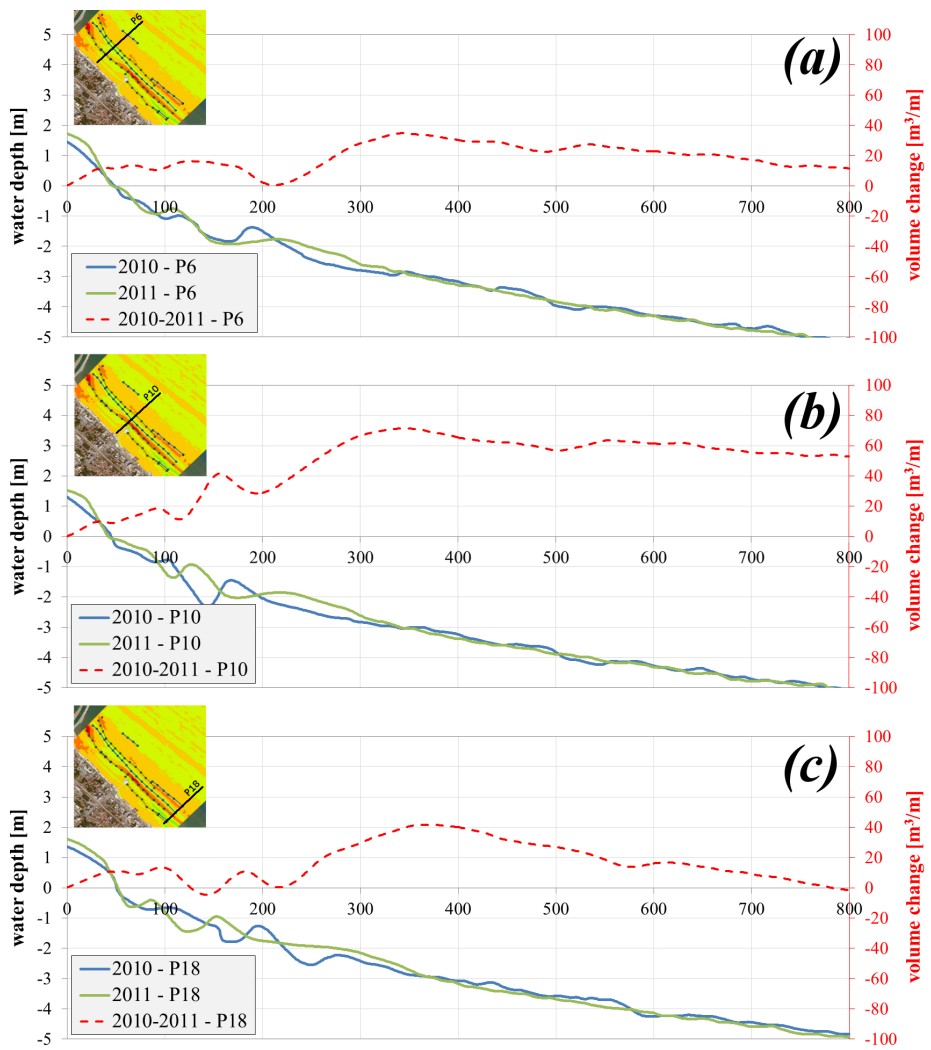

**Figure 6.** Cross-shore profiles collected in 2010 (solid blue lines) and 2011 (solid green lines), and cumulative volume variations in 2010-2011 (dashed red lines). The represented sections refer to **(a)** the Northern region, i.e. between the jetty and the "Rotonda", **(b)** the middle region, i.e. close to the "Rotonda", and **(c)** the Southern region.

the observed sediment transport, which is characterized by both cross-shore and alongshore contributions, significantly evolves and the closure depth increases, in agreement with the fitting depth increase nearby the structure (see Sect. 2).

The inspection of the cross-shore profiles and volume changes referring to the other time intervals confirms the existence of the three above-mentioned regions and, far from the "Rotonda", the inner and intermediate bars mainly contribute to the volume change in the upper beach and nearshore area. Further, close to the permeable structure, the volume changes more in 2011-2012 and less in 2012-2013 (e.g., see the regularity of both intermediate bar crest alignment and relative seabed variation in Fig. 4d). Finally, in 2010-2011 the balance throughout the profile, i.e. $VC|_{x=800m}$, is small far from the structures (Fig. 6a, c) and large in correspondence to the "Rotonda" (Fig. 6b), suggesting an important alongshore sediment transport localized nearby the structure. On the other hand, in 2011-2012 the farthest profiles (e.g., profile 18) are almost in equilibrium ($VC|_{x=800m} \sim 0$), while the alongshore contribution is important between jetty and "Rotonda", promoting an overall beach erosion ($VC|_{x=800m} \ll 0$). The $VC$ observed in 2012-2013 suggests a slight alongshore contribution throughout the domain.

## 4.3 Bar characterization

The previous data have been used to introduce a detailed analysis of the nearshore morphodynamics, especially the bar geometry and migration. Dimensionless parameters are introduced to analyze the bar geometry (e.g., see Grunnet and Ruessink, 2005). In Fig. 7a, the dimensionless bar height $H_{\mathrm{bar}}/h_{\mathrm{cr}}$ is plotted against the dimensionless bar width $W_{\mathrm{bar}}/s_{\mathrm{cr}}$, where the bar dimensions are defined as:

$$H_{\mathrm{bar}} = h_{\mathrm{tr}} - h_{\mathrm{cr}}, \tag{4}$$

$$W_{\mathrm{bar}} = 2(s_{\mathrm{cr}} - s_{\mathrm{tr}}). \tag{5}$$

In general, the bar height seems to increase with the bar width, this occurring for inner (+), middle (□) and outer (○) bars. Accounting for the surveys referring to 2010, 2011 and 2013, the outer bars are characterized by similar dimensionless heights ($H_{\mathrm{bar}}/h_{\mathrm{cr}}$ ranging between 0 and 0.26), but fairly different widths, the mean $W_{\mathrm{bar}}/x_{\mathrm{cr}}$ being of about 0.17 in 2010, 0.48 in 2011, 0.35 in 2013. The intermediate bars show similar trends, with $H_{\mathrm{bar}}/h_{\mathrm{cr}} = 0.35 - 0.4$ in 2010, 2011 and 2013, and $W_{\mathrm{bar}}/x_{\mathrm{cr}}$ significantly increasing in 2011 (0.54) and 2013 (0.54), with respect to 2010 (0.37). The 2006 middle bar behaves similarly to the 2013 middle bar, while the 2012 bars are always smaller in both height and width, as a consequence of the depth variations occurred in the preceding period. Hence, few and significantly small values referring to 2012 confirm the beach flattening occurred during the 2011–2012 period, dominated by Bora winds, which led to a general beach flattening, as already observed in Fig. 2 and Fig. 4c. No significant trends can be obtained from the inner bar data.

The analysis of the longshore distribution of the bar geometry can be undertaken accounting for the bar cross-shore area

$$\Omega = \frac{H_{\mathrm{bar}} W_{\mathrm{bar}}}{2}, \tag{6}$$

which is made dimensionless using both depth and distance to shore of the bar crest.

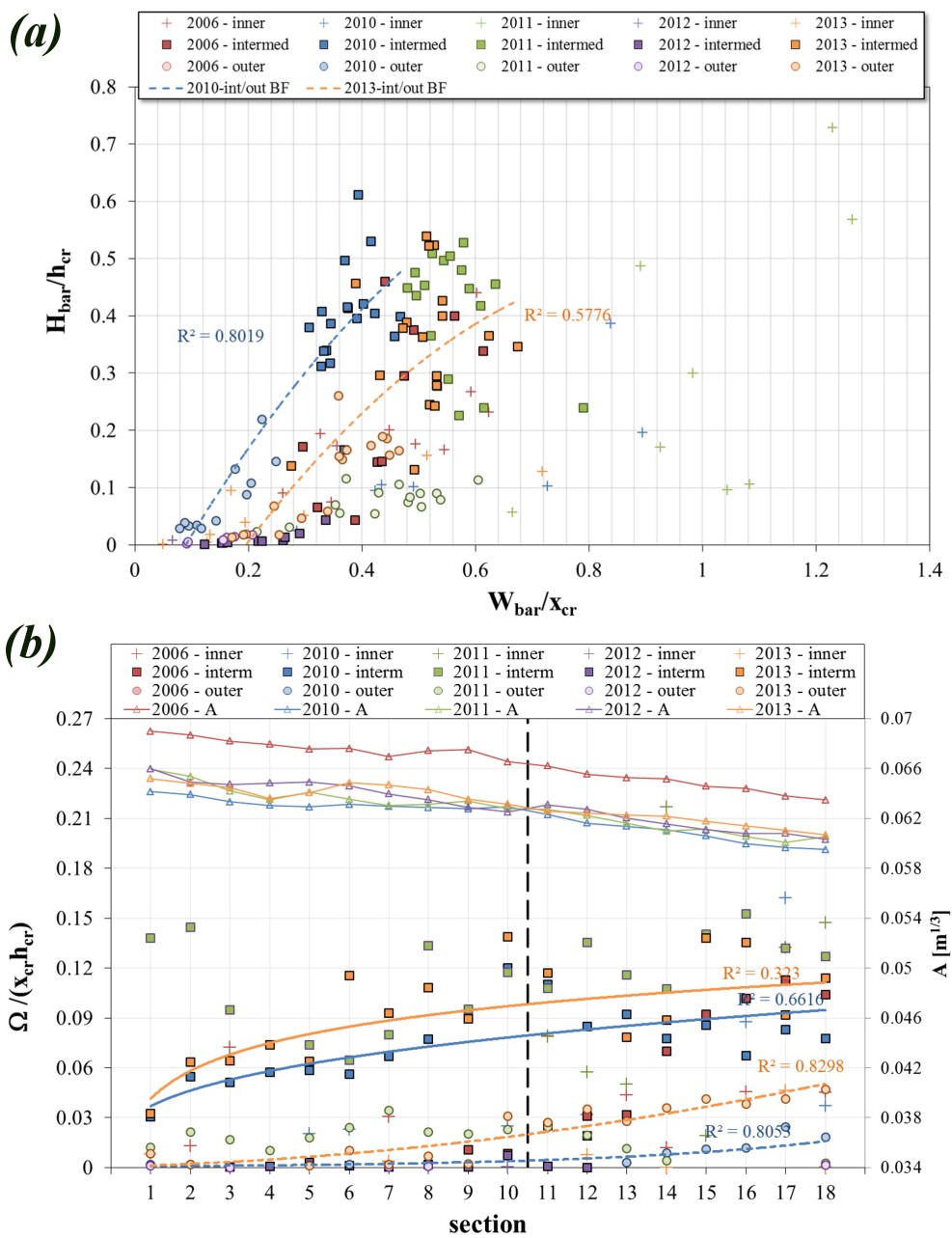

**Figure 7.** Dimensionless bar features: (a) bar height against bar width, (b) longshore distribution of bar cross-shore area (dashed and solid thick lines represent best-fit curves) and *A* parameter (solid thin lines and triangles). The vertical black dashed line represents the "Rotonda" location.

Figure 7b illustrates that, in general, all bars increase in dimension quite regularly moving southward. Focusing on years 2010, 2011 and 2013, the middle bars increase regularly between profiles 1 and 10, while South of the "Rotonda" (profiles 10–11), the trend is not clear. The outer bars seem to be more regular and aligned alongshore, and keep increasing moving southward. In 2006 the middle bar generates and starts increasing from profile 10, while in 2012 the trend is unclear, due to the reduced number of sections at which bars occur. In addition, it can be noticed that the shape parameter $A$ of Eq. 1 (solid thin lines and triangles), always decreases from left to right, suggesting a sediment-size reduction moving southward.

Hence, though Figure 7a illustrates a natural data scattering due to the beach variation in both time and space (e.g., see Grunnet and Ruessink, 2005), a best-fit polynomial curve well represents the geometrical characterization of outer and middle bars of 2010 (blue lines) and 2013 (orange lines). Further, Figure 7b shows that best-fit curves well reproduce the increasing trend of the outer bars moving southward ($R^2 > 0.8$ for 2010 and 2013, dashed lines), more than that of the middle bars ($R^2 > 0.3$, solid lines). The geometrical features of the inshore bars do not offer significant trends. In 2006, 2011 and 2012, weaker trends of the data represented in both panels are found for intermediate and outer bars, hence they have not been represented. This means that the inner bars are not characterized by homogeneous alongshore distributions, while middle and outer bars evolve locally, close to the structures, but only during specific time periods (e.g., see the 2011 intermediate bar at profiles 1 and 2, Figure 7b).

## 4.4 Bar dynamics

As suggested by several studies, the generation of subtidal bars may depend on three different mechanisms, i.e. i) breakpoint-related, ii) infragravity-waves-related, and iii) self-organisational mechanisms (e.g., see Wijnberg and Kroon, 2002; Leont'ev, 2011). From the results presented in Sect. 4.1 and 4.3, the bar dynamics in this area might be influenced by either the first or the second mechanism, while the self-organisation seems negligible. In fact, in agreement with Wijnberg and Kroon (2002), such a mechanism cannot explain the bar re-generation between 2012 and 2013 (Fig. 4d), after a general beach smoothing and the partial bar destruction occurred in 2011–2012 (Fig. 2 and Fig. 4c).

The destructive nature of the NNE storms significantly affects the bar geometry (beach smoothing), as well as the migration (seaward rather than shoreward), this being strongly influenced by the different wave features (waves coming from NNE were higher and steeper than those coming from ESE), which force the breaking to occur at different locations. Hence, the difference in terms of characteristics of the incoming sea-storm waves directly reflects on the beach morphology, underlining that the medium-term bar dynamics in the Adriatic sandy beaches are mainly governed by wind waves and breakpoint mechanisms.

Furthermore, steep NNE waves are associated with not excessive storm surges, while less steep ESE waves are associated with larger surges, due to the larger fetch which characterizes this wave direction in the Adriatic Sea. As an example, two consecutive intense storms occurred in December 2010, one coming from ESE, the other from NNE, were characterized by maximum surges of, respectively, $80$ and $43\,\mathrm{cm}$, measured within the protected basin of the Ancona harbor (data from Rete Mareografica Nazionale, ISPRA, http://www.mareografico.it). This leads to larger water depths over the crest ($h_{\mathrm{cr}}$) and smaller relative wave heights ($H/h_{\mathrm{cr}}$) during ESE than during NNE waves. In fact, wave propagation from the offshore to the outer bar depth enables one to estimate the local wave inclination ($\alpha_l$), and also the local wave height $H_{\mathrm{m0},l}$. This may be done using

either simple analytical models, which accounts for wave refraction and shoaling (e.g., Goda, 2000), or detailed numerical approaches, which provide a more complete wave characterization in the nearshore (e.g., Carniel et al., 2011). Then, the actual water depth over the crest during surge may be estimated as $h_{cr,s} = \overline{h_{cr}} + \eta_s$, where $\overline{h_{cr}}$ is the alongshore-averaged still water depth over the crest and $\eta_s$ the surge contribution, which is different depending on the dominating wave direction, but in

agreement with both the data collected at the Ancona harbor (Rete Mareografica Nazionale, ISPRA) and previous literature studies (Orlić et al., 1994; Villatoro et al., 2014).

The above-introduced terms and the relative wave height estimated using the local root-mean-square wave height $H_{\mathrm{rms},l}$ (US Army, 1977) are summarized in Tab. 3. The local values have been obtained transferring the offshore wave to a finite depth of $3m$, i.e. just off the outer bar locations, using Goda (2000)'s approach. The relative wave height, especially $H_{rms,l}/h_{cr,s}$, which

is larger in 2010–2011 and 2012–2013 and smaller in 2011–2012, suggests, respectively, a landward and seaward bar migration, which has been actually observed (e.g., see Fig. 4). The estimated local wave angles suggest an almost orthogonal-to-shore direction during the NNE-wave-dominated period. Our observations are supported by the numerical results of Dubarbier et al. (2015), who found that the variability in sandbar migration is sensitive to water level over bar crest, being consistent with storm-surge variations occurring in our site. On the other hand, wave obliquity mainly affects the rates of bar growth and

migration, but not their migration direction. This suggests that the difference between Bora and Scirocco waves, in terms of wave incidence, does not influence the bar direction, but eventually their propagation speed. The outer bar variation, i.e. the change of the alongshore-averaged outer bar height ($\Delta \overline{H_{\mathrm{bar}}}/h_{cr,s}$) and cross-shore area ($\Delta \overline{\Omega}/(\overline{x_{cr}} h_{cr,s})$), as well as the outer bar migration ($\Delta \overline{s_{cr}}$), has also been analyzed. As illustrated in Tab. 3, in 2010-2011 the outer bar, which globally moves shoreward, slightly reduces in height and increases in area. In 2011-2012 the bar, which moves seaward, significantly reduces

in height and area. Conversely, in 2012-2013 the bar largely increases in height and area, regenerates and moves shoreward. It is worth noting that the smallest changes occur in 2010-2011, when a double peak characterizes the wave climate (Fig. 3b), while significant changes occur during the following intervals, when the climate is clearly dominated by NNE (2011-2012, bar decrease) or by ESE (2012-2013, bar increase) forcing. Further, the outer bar migration in 2010-2011 and 2012-2013 is fairly similar, i.e. $\sim 63m$, providing an migration rate of $(0.17-0.18)\,\mathrm{m/day}$, while in 2011-2012 the outer bar migrates seaward

with a rate of $\sim 0.26\,\mathrm{m/day}$, such values being in agreement with typical literature findings (e.g., van Enckevort and Ruessink, 2003).

**Table 3.** Estimate of relative wave height, wave incidence and outer bar-geometry change for the examined time periods.

| Time range | $H_{\mathrm{m0}}$ | $H_{\mathrm{m0},l}$ | $H_{\mathrm{rms},l}$ | $\overline{h_{cr}}$ | $\eta_s$ | $h_{cr,s}$ | $H_{\mathrm{m0},l}/h_{cr,s}$ | $H_{\mathrm{rms},l}/h_{cr,s}$ | $\alpha_l$ | $\Delta\overline{H_{\mathrm{bar}}}/h_{cr,s}$ | $\Delta\overline{\Omega}/(\overline{x_{cr}}h_{cr,s})$ | $\Delta\overline{s_{cr}}$ |
|---|---|---|---|---|---|---|---|---|---|---|---|---|
| [years] | [m] | [m] | [m] | [m] | [m] | [m] | [-] | [-] | [°] | [-] | [-] | [m] |
| 2010–2011 | 1.75 | 1.51 | 1.07 | 2.33 | 0.60 | 2.93 | 0.52 | 0.36 | 22 | -0.016 | 0.787 | 68.63 |
| 2011–2012 | 2.25 | 2.48 | 1.75 | 1.81 | 0.35 | 2.16 | 1.15 | 0.81 | 5 | -0.050 | -1.335 | -112.29 |
| 2012–2013 | 1.75 | 1.39 | 0.98 | 2.75 | 0.60 | 3.35 | 0.41 | 0.29 | 24 | 0.063 | 1.796 | 68.82 |

## 5   Discussion

Recent studies on the dynamics of barred beaches led us to correlate wave-climate and bathymetric surveys of an unprotected beach of the Adriatic Sea. In fact, though some results on sandbar migration along the Tyrrhenian Sea were recently illustrated (e.g., Parlagreco et al., 2011), the bar dynamics of typical Adriatic sandy beaches have not been already investigated. Further, the correct understanding of the bar migration is important when dealing with beach management and tourism. To this aim, the coast of Senigallia has been here investigated since, similarly to many Adriatic sandy beaches, this is characterized by a significant flow of tourism, especially in the summertime (see Sect. 2).

Hence, the bathymetric surveys of the area South of the harbor, which has been seen to be stable in the long term, enabled us to analyze a multiple-bar array typical of the sandy beaches of the Middle Adriatic. Such a part of the basin is subject to sea storms mainly due to NNE (Bora) and ESE (Levante-Scirocco) winds, which are characterized by significantly different surges.

The seabed-depth variation and the wave climate between consecutive surveys, as well as the bar features (height, width, location) analyzed for each survey, enabled us to couple the beach/bar dynamics with the wave forcing.

In the studied area the tidal excursion ($\sim 40\,\mathrm{cm}$) is small and only subtidal bars exist. Since the analyzed beach slope ranges between $1:35 \sim 0.03$ (swash zone) and $1:200 \sim 0.005$ (offshore area), such bars fall into the group of two-dimensional longshore bars (Wijnberg and Kroon, 2002). Further, the wave energy in such a microtidal environment is quite high.

In the analyzed region and during the investigated time periods, the beach experienced many sea storms that enabled us to give an overall interpretation to the bar migration process as a function of the wave climate. Coupling wave steepness and the Dean number (i.e. the ratio of wave height to sand fall velocity and wave period), both ESE and NNE are associated with erosive wave conditions (e.g., see Dean and Dalrymple, 2004). However, during the time periods dominated by ESE forcing, waves are characterized by a reduced steepness $H_{\mathrm{m0}}/L_{\mathrm{p0}} = 0.213$ (exactly the same in 2010–2011 and 2012–2013), while this is about $1/3$ larger during the NNE-forcing-dominated period ($H_{\mathrm{m0}}/L_{\mathrm{p0}} = 0.316$). Such a behavior is confirmed if we do not account for the most energetic waves (see Sect. 4.1), but directly estimate the most frequent combination ($H_{\mathrm{m0}}$, $T_{\mathrm{p}}$). Further, an increase of the bar steepness $H_{\mathrm{bar}}/W_{\mathrm{bar}}$ is associated to a decrease of $H_{\mathrm{m0}}/L_{\mathrm{p0}}$ (e.g., compare the bar geometry in Fig. 2 with the associated wave steepness).

As already stated, steep NNE waves, associated to reduced storm surges, lead to larger relative wave heights $H/h_{cr,s}$, while less steep ESE waves lead to smaller values. As observed by Houser and Greenwood (2005), relative rms heights $H_{\mathrm{rms},l}/h_{cr,s} = 0.3 - 0.4$ lead to a landward bar migration, associated with bar height increase. This occurs for the outer bar in 2012–2013 ($H_{\mathrm{rms},l}/h_{cr,s} = 0.29$), when the bar is almost completely regenerated (see also Fig. 2), but not between 2010–2011 ($H_{\mathrm{rms},l}/h_{cr,s} = 0.36$), when the bar height slightly decreases (see Tab. 3). Conversely, values of $H_{\mathrm{s},l}/h_{cr,s} > 0.6$ lead to a seaward bar migration, as observed in 2011–2012 ($H_{\mathrm{rms},l}/h_{cr,s} = 0.81$), when the outer bar is partially destroyed and its height significantly decreases. Further, waves coming from ESE are characterized by a significant longshore component, due to the large angle between the approaching wave fronts and the coast (see Tab. 3). Differently, waves coming from NNE reach the shore with an almost perpendicular incidence, improving the intense smoothing of the bars.

Hence, it has been seen that the relative wave height can be properly applied for the prediction of bar migration in an environment different from those already proposed in the literature (e.g., Ruessink and Terwindt, 2000; Houser and Greenwood, 2005), i.e. a nearshore area characterized by a reduced tidal excursion, and partially influenced by the presence of rigid structures. This allows the application of such a predictive parameter for similar nearshore environments, and also for a medium-term predic-
tion. Hence, such a parameter is valid for different environments, characterized by tidal excursions of some centimeters (e.g., Lake Huron, Houser and Greenwood, 2005) to decimeters (Adriatic Sea, present study) to meters (e.g., North Sea, Ruessink and Terwindt, 2000). Assuming that the bar migration mainly occurs during sea storms, the involved sediment transport mainly depends on the incoming short waves (especially when the bars move landward, i.e. ESE waves dominating) and the undertow (especially for seaward motion, associated with NNE waves), with the infragravity waves probably being of some importance
in such a dissipative beach (e.g., see Wright and Short, 1984; Ruessink et al., 1998).

While the correlation between bar width and bar height is clear only for some cases, the width increasing with the height, an overview of the available data enable further conclusions. Between 2010 and 2011, the largest waves, mainly propagating from ESE, provided a height increase of the outer bar (in agreement with Houser and Greenwood, 2005) only North of the "Rotonda", and, at the same time, a width increase and a steepness reduction of both outer and intermediate bars (blue and
green symbols in Fig. 7a). While between 2011 and 2012 the bars are largely smoothed due to the NNE dominating waves (purple symbols), the ESE stormy conditions occurred between 2012 and 2013 gave rise to geometric features of the bars similar to those observed in 2011 (orange symbols).

The cross-shore bar area increases moving southward, especially from the Senigallia harbor to the "Rotonda", which partially disturbs the growth of the middle bar. This could also be analyzed in view of the equilibrium-profile theory, described by Eq. 1.
The analysis of the shape parameter $A$ (see Fig. 7b) suggests that $d_{50}$ slightly decreases moving southward. Some important oscillations of $A$ characterize the region between profiles 1 and 11, underlining the influence of the rigid structures, while a generally decreasing trend can be observed South of the "Rotonda" (notice that the larger values referring to 2006 may be due to the lower resolution of the surveyed bathymetry in the nearshore , i.e. up to a depth of $1.5 - 2m$, with respect to the following surveys). Such a decrease is in agreement with the sediment-size distributions observed in 1989 and 1990 by
Lorenzoni et al. (1998a). It is worth noting that the rigid structures directly affect the sediment transport and generate complex hydro-morphodynamics, which, in turn, influence the bed morphology and bar geometry.

This is probably due to: i) the river jetty (Fig. 1a), which induces a complex flow field, i.e. a mix of refraction, diffraction and reflection, that generates wave-wave interactions, crossing waves and intense vorticity, especially when sea storms come from ESE (e.g., see Postacchini et al., 2014); ii) the river discharge, especially during severe weather conditions, which gives rise
to an intense plume that both propagates southeastward and promotes sediment deposition along its path (e.g., see Brocchini et al., 2017). Hence, the dynamics induced by such phenomena suggest both a deposition of larger sediments immediately south of the jetty, where a more turbulent flow field exists, and a mobilization of finer sands coupled with their transport far from the jetty.

The similar geometry of the bars (width, height, steepness, cross-shore area) in 2011 and 2013, hence suggests that sim-
ilar medium-term wave features (direction, height, period in 2010-2011 and 2012-2013, respectively) provide similar beach

responses, while the initial morphological conditions, respectively represented by the 2010 and 2012 surveys, though significantly different, slightly affect the beach evolution. Further, permeable and impermeable structures locally affect the dynamics of the submerged bars, but do not change their migration direction and their macroscopic features, which are thus dominated by the dominant wave forcing. In addition, the outer bar significantly changes when the climate is clearly dominated by either NNE (2011-2012) or ESE (2012-2013) waves, increasing/decreasing during NNE/ESE forcing. Conversely, small changes (cross-shore area increase and height reduction) occur in 2010-2011, when the wave climate is not clearly dominated by NNE or ESE waves.

## 6 Conclusions

The nearshore dynamics is characterized by different levels of analysis: i) the long-period beach stability is of the order of decades, ii) the medium-term evolution of the beach forms (e.g., submerged bars, artificial nourishments) is of the order of years or seasons, and iii) the short-term erosion of the beach profile is of the order of days or hours. While i) and iii) have been widely investigated, the medium-term beach variability has not been sufficiently analyzed. Hence, recent findings led us to investigate the medium-term morphodynamics of the sandy barred beach of Senigallia, located in the Middle Adriatic Sea.

The present work both illustrates how a proper buoy-data handling leads to the prediction of the morphological changes of a barred beach and offers a useful tool, for coastal engineers and managers, to: i) properly predict the emerged beach stability (e.g., shoreline retreat, erosion), ii) accurately design nourishments for submerged beach recovery, iii) estimate the sediment transport flux through the entrance of nearby harbors, iv) choose the best place to drop the dredged sediment coming from nearby harbors, eventually with nourishment purposes.

A more complete analysis could be achieved through use of either data collected by another waverider (e.g., that of Cesenatico, FC, which is $\sim 80\,\mathrm{km}$ North of Senigallia) or model hindcasts (e.g., Mentaschi et al., 2015), with the aim to characterize the wave forcing during further time periods, after a proper data validation. Although global reanalysis or numerical modeling may provide a more detailed wave characterization, use of available regional climate models (e.g., The Medatlas Group, 2004) is easier and may represent a valid alternative. Further, the dynamics of the nearshore area before, during and after storm events could also be inspected by means of novel devices like: i) Lagrangian drifters, capable of measuring both three-dimensional hydrodynamics and seabed depth (e.g., Postacchini et al., 2016a), ii) video-monitoring systems, like that available at the Senigallia harbor since 2015, to reconstruct the coastline (e.g., Archetti, 2009; Vousdoukas et al., 2011; Archetti et al., 2016), as well as wave field and bed morphology (e.g., Palmsten et al., 2015), iii) radar images, like those used for the reconstruction of both wave field and bathymetry, through the depth inversion technique (e.g., Ludeno et al., 2015).

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
