# Peer review of "Medium-term dynamics of a Middle Adriatic barred beach"

_Ocean Science, 2016_

## Referee Comment (RC1) · Anonymous Referee #1 · 7 Feb 2017

General comments

The paper focuses on the morphodynamic analysis of a natural sandy littoral stretch located along the highly urbanized coastal sector of the Western Adriatic Sea. Morphologic variability of submerged sandbars is quantitatively analysed and compared with in situ wave data to evaluate the near-shore medium-term dynamics. The study domain is located in a semi-enclosed and elongated basin whose coastal dynamics are deeply influenced by the physiographical setting. The paper furnishes new insight on local (Central Adriatic) and general (semi-enclosed basin) bars behaviour in medium-term. For this reason, and with the aim to extend the local findings to more general ones, the authors use proper conceptual and analytical methods to describe data and results. However, in order to fully constrain the reached conclusions, a better description of some evidences on the sandbars morphologic variability is needed.

[Figure]

Specific comments

"Sandbar cross-shore migration evidences"

In the section 4.2 "bathymetric surveys", the proposed seaward/shoreward sandbars migration patterns between consecutive surveys (2006, 2010, 2011, 2012, 2013) are not clearly detectable. Considering the importance of this phenomenon in relation to the paper objective, a better description and presentation are required. Referring to Figure 5, some considerations are reported below. The 2006-2010 shoreward migration is highly localized around the "rotonda sections", the southernmost transects on the contrary seem not to highlight any kind of real net cross-shore displacement. The 2010-2011 evolution is represented by a detectable sandbars cross-shore displacement but not uniform alongshore. The Authors propose a shoreward migration, thus assuming that the sand volume stored in 2010 on the middle-bar had contributed to the upper profile evolution and the large sand volume stored in 2011 on the outer-bar would be alongshore derived (see for example Figure 2). On the contrary, a seaward displacement would imply that the sand volume stored in 2010 on the middle-bar had contributed to the lower profile evolution and the net alongshore sand movement would be located in shallower depth. Even if this is a very limited example, qualitatively detected by a single transect, a volumetric approach could be fundamental to evaluate the "cross-transect and along-transect" sedimentary mobility, thus improving the alongshore consistency of interpreted cross-shore sandbars displacement.

"Sandbar alongshore variability"

As stated by the Authors, since the area is located close to the jetty and characterized by complex hydrodynamics, it could be likely to develop complex sandbar morphologies. Moreover during particular storm events, the coupling of complex hydrodynamics and morphologies could induce cross-shore beach profile response, not necessary in phase with areas farther away from "jetty-rotonda" (see for example Shand et alii, 2001), thus the generalization on the bars behaviour along the whole area could be

more complex to define. In this framework, as noted by the Authors, the stabilization of the "bar features" along the southernmost transects could testify the boundary between the near-shore area influenced and "not directly influenced" by the jetty and the "rotonda" structures. The same applies to the cross-shore limits of "rotonda" influence on sandbar characteristics, as well evidenced and described in Figure 5b.

References

Shand, R., Bailey D. and Shepherd, M. (2001) – Longshore realignment of shore-parallel sand-bars at Wanganui, New Zealand. Marine Geology, 179, 147-161.
* * *

---

## Referee Comment (RC2) · Anonymous Referee #2 · 7 Feb 2017

**General Comments**

Summary: This submission investigates medium-term morphodynamics of a barred beach along the Middle Adriatic coast of Italy using annual bathymetric surveys and offshore wave buoy data. Previous studies of similar beaches have focused on shortand long-term dynamics, leaving medium-term behavior relatively unstudied. A better understanding of the connections between wave climate and changes in nearshore morphology is needed to improve models that predict storm effects (flooding), beach erosion, or how shoreline protection structures will affect a beach. The authors utilize well-tested data sources (bathymetric surveys and wave buoy from Italian wave measurement network) to examine dynamics on a neglected timescale, medium-term. However, the bathymetric data was collected only once per experimental year, and in my opinion, this limited data set imposes restrictions on the authors' interpretation of

the results. Namely, there is insufficient data to separate the possible effects of shortterm changes due to winter storms from medium-term changes due to medium-term wave climate. See Specific Comments section for further comment.

Abstract and Title: Abstract explains context and main findings clearly and concisely. The title is appropriate, although upon first reading, the phrase "mediumterm dynamics" was unfamiliar. After reading the abstract, I learned that mediumterm is a timescale that is longer than short-term (days) and shorter than long-term (years/decades), on the order of seasons or annual. If there is a way to be more clear in the title that medium-term is a timescale, that would be encouraged, but this change is not essential.

Organization: This paper is generally well organized. One recommendation- Explain Dean-type beach stability analysis earlier. It is not addressed until the discussion, but it is first mentioned in the "Description of the Site" and then used to explain longshore variability in the "Results". It would be helpful to present the equation with its explanation earlier, so the reader knows where this stable beach shape characterization comes from and why it is relevant for understanding other changes to the beach morphology.

**Specific Scientific Comments**

Assessment of Medium-term dynamics: Wave climate and nearshore morphology are strongly linked and it is valuable to reveal this relationship over various time scales and environments. The authors focus on a sandy barred beach, chosen for its similarity to many other beaches worldwide, over medium-term time scales. The wave data presented here is sufficient for medium-term analysis. However, a description of the original form of the wave buoy data (time series, hourly product, wave spectra?) and the methods used to further process this data should be specified for the sake of reproducibility.

The authors acknowledge that beach morphology is "dynamic throughout the year, especially during sea storms driven by NNE winds" (p. 3, line 30). Given this variability, it

OSD
is necessary to be able to distinguish short-term variability from medium-term variability in order to assess the connection between medium-term wave climate and beach morphology. The bathymetric surveys were collected in different months/seasons each year: June/Summer 2006, February/Winter 2010, February/Winter 2011, April/Spring 2012, March/Spring 2013. The authors explain that the two types of winds (ESE and NNE) can happen during the same season, and that in the study region, winters are stormy and summers are calm. The winter surveys would therefore be more susceptible to short-term variability due to storms, which is not distinguishable given the limited bathymetric data set. The summer bathymetric surveys are perhaps more representative of medium-term dynamics, because the beach is not subjected to the magnitude or frequency of short-term, high impact events during that season. Since the literature shows and the authors admit that significant bathymetric changes can occur over the course of a single storm, and there is no information given to put each survey in this short-term context, it is not correct to assume that the "snapshot" of bathymetry seen in the data is representative of the medium-term dynamics. The authors should pursue supplementary data (perhaps short-term wave data analysis) to provide context for the bathymetry surveys used in this study.

The authors present current theory, based on peer-reviewed and published field and laboratory measurements, for predicting bar migration based on wave conditions. This theory states that steeper, larger waves promote a seaward shift of the bar and less steep, smaller waves promote a shoreward shift of the bar. The bar migration pattern results presented in this study agree with previous findings. However, agreement with the theory is limited due to insufficient bathymetric data to definitively ascribe morphological changes over a particular year to medium-term wave climate alone (i.e., lacking evidence that short-term wave climate is not contaminating the bathymetric surveys).

Tables: Table 1& 2: The authors claim it is best to use wave statistics based on the maximum percentage of energy flux over the time interval of interest. This is a fair decision. However, based on the Figure 3 wave roses and Table 1 statistics, there
is not always a single band where the energy flux is concentrated. For 2010-2011 especially, the energy flux seems evenly split between H\_m0=1.5-2.0 m (energy flux distribution % of 16.56) and H\_m0=3.0-3.5 m (energy flux distribution % of 16.02). The authors decide that the dominant waves were about Hm0=1.75 m. Looking at the wave rose for 2010-2011 there are strong peaks in both the ESE (~25%) and NNE (~20%) directions. Yet, when summarizing the 2010-2011 period, the authors choose ESE for this time interval. The 2011-2012 and 2012-2013 conditions were truly dominated by one type of wind event over the other, so the assumptions made by the authors for those time intervals are justified. Perhaps a more nuanced bimodal analysis of 2010-2011 is warranted, especially if these bulk wave climate characteristics are used to explain changes in beach morphology.

Figures: Figure 6a shows normalized bar height versus normalized bar width with fits for outer and inner bar (essentially steepness curves (H\_bar/W\_bar) showing how bar geometry changes from outer to intermediate bars). The fits are presented for 2010 and 2013, but not for the other three years of data, leaving the reader questioning whether these trends are consistent. Furthermore, if the goal is to show that medium-term bar dynamics are strongly linked to medium-term wave climate, it is important to present plots that relate bar features (or changes in bar features) to wave climate metrics (like Table 3, but is visual plot form). Figure 6b shows that the cross-shore area of the bar increases southward. A shift in the grain size distribution is the explanation given for the alongshore trend in the equilibrium beach profile. Since grain size distribution is a consideration throughout the authors' analysis of the results, plotting cross-shore bar area versus some grain size distribution metric would be more useful.

**Technical Comments**

Fluency: Although it is apparent that English is the second language of the authors, this does not inhibit the reader's ability to understand this research and its conclusions. There are only a few places where grammar issues impede the authors' message. Listed below are the sentences where a second pass at phrasing would be beneficial.
p. 7, line 10 - p. 8, line 3 p. 16, lines 1-7 p. 18, lines 15-20

Equations: Mathematical formulae, symbols, abbreviations, and units are correctly defined and used.

References: The number and quality of references is appropriate.

---

## Referee Comment (RC3) · Anonymous Referee #3 · 9 Feb 2017

Paper OS-2016-106

I have read with interest the paper "Medium-term dynamics of a middle Adriatic barred beach" by Postacchini et al. The MS paper deals with the morphodynamic analysis of a natural sandy littoral stretch located along a highly urbanized coastal sector on the west Adriatic Sea. The morphologic variability of submerged sandbars is analysed and compared with in situ wave data, on order to evaluate medium-term dynamics.

Generally speaking, I find the paper interesting, tackling an important aspect in a convincing way (although of course dealing with the usual problems of having "not enough" data) and well organized. The English is also rather fluent. However, the MS may benefit from some minor improvements that could be easily implemented by the authors.

-I feel the need of more specific links to some of the several existing wave-climate stud-

ies on the Adriatic basin. - The paper tackles a subject relatively unexplored, i.e. the medium-term behavior, that is a timescale that between short term (days) and long term (years/decades), on the order of seasons or years, using annual bathymetric surveys and offshore wave buoy data. There is no doubts that improving the knowledge on the relations between wave climate and changes in nearshore morphology is a necessary step, in order to improve numerical models capable of predicting storm effects, beach erosion, and more generally the efficiency of shoreline protection measures. This is even more valid in a context of climate change, that should be also mentioned in the paper more clearly. Benetazzo et al. 2012, DOI:10.5194/nhess-12-1-2012, and references therein inlcluded may give some useful hints on this.

-at the same time, some lines addressing the realtionship between the local scale dynamics with a more regional scenario of sediment dynamics and transport should possibly be introduced by the authors. Sherwood et al., 2004, Oceanography; or Harris et al., DOI: 10.1029/2006JC003868. Even though not strictly pertinent to the study, it should be indeed mentioned that the longshore and cross-shore budget of the local beach is however to be framed within a more regional dynamics.

-other existing approaches could be mentioned in order to provide a more complete series of wave data nearshore, including the transfer of offshore wave data to the coast by means of wave models, e.g. SWAN, in order to reconstruct a more detailed and spatially meaningful wave climate (Carniel et al., 2011, DOI: 10.2478/s13545-011-0036-1)

-Some more caveats should be discussed, since the bathymetric data was collected once per year and are therefeore somehow limited. Possible workarounds could be put more in evidence, as the use of video images to reconstruct the coastline (see Archetti et al., 2016, . doi:10.5194/nhess-16-1107-2016, and references therein included

-Since it is somehow difficult to be sure about the separation of short-term changes due to winter storms with respect to medium-term changes due to medium-term wave climate, in this filed even a relatively quick reference/analysis of wind and wave data

resulting from climate models or satellite-verified database may be of direct help. Although the wave data presented here seems to be sufficient for the proposed medium-term analysis, and a more careful analysis of available wave-data also from global reanalysis or modeling efforts would improve the soundness of analysis but being too heavy, the authors may refer to existing efforts that are represented by available regional cliamte models, originated possibly from efforts such as the MEDATLAS project http://users.ntua.gr/mathan/pdf/Pages_from%20_WIND_WAVE_ATLAS_MEDITERRANEAN_SEA_2004.pdf

---

## Author Comment (AC1) · 1 Mar 2017

**We thank Reviewer $\#1$ for their useful comments and suggestions which will help to improve our manuscript. The comments from the Reviewer below are in italic font and our point-by-point responses are in bold.**

*General comments*
*The paper focuses on the morphodynamic analysis of a natural sandy littoral stretch located along the highly urbanized coastal sector of the Western Adriatic Sea. Morphologic variability of submerged sandbars is quantitatively analysed and compared with in situ wave data to evaluate the near-shore medium-term dynamics. The study domain is located in a semi-enclosed and elongated basin whose coastal dynamics are deeply*

*influenced by the physiographical setting. The paper furnishes new insight on local (Central Adriatic) and general (semi-enclosed basin) bars behaviour in medium-term. For this reason, and with the aim to extend the local findings to more general ones, the authors use proper conceptual and analytical methods to describe data and results. However, in order to fully constrain the reached conclusions, a better description of some evidences on the sandbars morphologic variability is needed.*

**We thank the Reviewer for their appreciation of our analysis. As suggested below, the new version of the manuscript will better describe the evidences of sandbar migration and morphological variability of the beach.**

*Specific comments*
*"Sandbar cross-shore migration evidences"*
*In the section 4.2 "bathymetric surveys", the proposed seaward/shoreward sandbars migration patterns between consecutive surveys (2006, 2010, 2011, 2012, 2013) are not clearly detectable. Considering the importance of this phenomenon in relation to the paper objective, a better description and presentation are required. Referring to Figure 5, some considerations are reported below. The 2006-2010 shoreward migration is highly localized around the "rotonda sections", the southernmost transects on the contrary seem not to highlight any kind of real net cross-shore displacement. The 2010-2011 evolution is represented by a detectable sandbars cross-shore displacement but not uniform alongshore. The Authors propose a shoreward migration, thus assuming that the sand volume stored in 2010 on the middle-bar had contributed to the upper profile evolution and the large sand volume stored in 2011 on the outer-bar would be alongshore derived (see for example Figure 2). On the contrary, a seaward displacement would imply that the sand volume stored in 2010 on the middle-bar had contributed to the lower profile evolution and the net alongshore sand movement would be located in shallower depth. Even if this is a very limited example, qualitatively detected by a single transect, a volumetric approach could be fundamental to evaluate*

*the "cross-transect and along-transect" sedimentary mobility, thus improving the along-shore consistency of interpreted cross-shore sandbars displacement.*

**Following Reviewer's suggestions, the description of the sandbar cross-shore migration will be improved by i) better illustrating Figs.4 and 5, ii) including new cross-shore profiles at different alongshore locations, to both clearly illustrate bar-migration evidence and highlight the influence of existing structures on morphological changes, iii) plotting and describing volume changes along different cross-shore profiles.**

*"Sandbar alongshore variability"*
*As stated by the Authors, since the area is located close to the jetty and characterized by complex hydrodynamics, it could be likely to develop complex sandbar morphologies. Moreover during particular storm events, the coupling of complex hydrodynamics and morphologies could induce cross-shore beach profile response, not necessary in phase with areas farther away from "jetty-rotonda" (see for example Shand et alii, 2001), thus the generalization on the bars behaviour along the whole area could be more complex to define. In this framework, as noted by the Authors, the stabilization of the "bar features" along the southernmost transects could testify the boundary between the near-shore area influenced and "not directly influenced" by the jetty and the "rotonda" structures. The same applies to the cross-shore limits of "rotonda" influence on sandbar characteristics, as well evidenced and described in Figure 5b.*

**About this point, the three-dimensional bar behavior and the influence of the existing structures will be better discussed. Following Shand et al. (2001)'s[1] findings, the three-dimensional patterns developing along the coast should be ascribed to complex hydrodynamics, hence not always to high-energy conditions,**
* * *
[1]Shand, R. D., Bailey, D. G., & Shepherd, M. J. (2001). Longshore realignment of shore-parallel sand-bars at Wanganui, New Zealand. Marine Geology, 179(3), 147-161.

**but also to other factors, like low-frequency waves and morphological variations. In addition, the above-mentioned cross-shore profiles and volume changes will be used to improve the discussion on such a point.**

---

## Author Comment (AC2) · 1 Mar 2017

**We thank Reviewer #3 for their precious suggestions, which will be implemented to improve the quality of our manuscript. The comments from the Reviewer below are in italic font and our point-by-point responses are in bold.**

*I have read with interest the paper "Medium-term dynamics of a middle Adriatic barred beach" by Postacchini et al. The MS paper deals with the morphodynamic analysis of a natural sandy littoral stretch located along a highly urbanized coastal sector on the west Adriatic Sea. The morphologic variability of submerged sandbars is analysed and compared with in situ wave data, on order to evaluate medium-term dynamics. Generally speaking, I find the paper interesting, tackling an important aspect in a convincing*

[Figure]

*way (although of course dealing with the usual problems of having "not enough" data) and well organized. The English is also rather fluent.*

**We thank again the Reviewer for their appreciation of our work.**

*However, the MS may benefit from some minor improvements that could be easily implemented by the authors.*
*-I feel the need of more specific links to some of the several existing wave-climate studies on the Adriatic basin. - The paper tackles a subject relatively unexplored, i.e. the medium-term behavior, that is a timescale that between short term (days) and long term (years/decades), on the order of seasons or years, using annual bathymetric surveys and offshore wave buoy data. There is no doubts that improving the knowledge on the relations between wave climate and changes in nearshore morphology is a necessary step, in order to improve numerical models capable of predicting storm effects, beach erosion, and more generally the efficiency of shoreline protection measures. This is even more valid in a context of climate change, that should be also mentioned in the paper more clearly. Benetazzo et al. 2012, DOI:10.5194/nhess-12-1-2012, and references therein inlcluded may give some useful hints on this.*

**We agree with the Reviewer. In the revised version of the manuscript, we will properly discuss the morphological changes of a barred beach, induced by the wave forcing, within the climate-change frame.**

*-at the same time, some lines addressing the realtionship between the local scale dynamics with a more regional scenario of sediment dynamics and transport should possibly be introduced by the authors. Sherwood et al., 2004, Oceanography; or Harris et al., DOI: 10.1029/2006JC003868. Even though not strictly pertinent to the study, it should be indeed mentioned that the longshore and cross-shore budget of the local beach is however to be framed within a more regional dynamics.*

**To better contextualize the observed morphological changes in the Adriatic framework, we will introduce the suggested regional aspects in Section 2.**

*-other existing approaches could be mentioned in order to provide a more complete series of wave data nearshore, including the transfer of offshore wave data to the coast by means of wave models, e.g. SWAN, in order to reconstruct a more detailed and spatially meaningful wave climate (Carniel et al., 2011, DOI: 10.2478/s13545-011-0036-1)*

**Numerical models commonly used to reconstruct the wave propagation from the offshore to the nearshore region will be mentioned, in addition to the classical Goda (2000)'s approach[1].**

*-Some more caveats should be discussed, since the bathymetric data was collected once per year and are thereofore somehow limited. Possible workarounds could be put more in evidence, as the use of video images to reconstruct the coastline (see Archetti et al., 2016, . doi:10.5194/nhess-16-1107-2016, and references therein included.*

**We agree with the Reviewer and their suggestions will be implemented. Further, a video-monitoring station has been recently installed at the Senigallia harbor to collect images of the coastal area between the jetty and the "Rotonda". Recently discussed preliminary data[2] seem to confirm our conclusions about bar migration. Such a system will be exploited to analyze the short-term bar migration in dedicated future works. Further, recent experiments carried out at the Misa River estuary identified an intense nearbed sediment transport towards the offshore just off the estuary[3]. This occurred during a NNE storm and suggested a**
* * *
[1]Goda, Y. (1985). Random seas and design of marine structures. Advanced series on ocean engineering vol. 15, World Scientific, Singapore.

[2]Palmsten M.L., Calantoni, J., Brocchini M., Soldini L. & Postacchini M. (2016). Sand bar behavior in a mixed sediment environment, Ocean Sciences Meeting. https://agu.confex.com/agu/os16/meetingapp.cgi/Paper/89790

[3]Brocchini M., Calantoni J., Postacchini M., Sheremet A., Staples T., Smith J., Reed A.H., Braithwaite III E.F.,

**seaward bar migration in the studied area.**

*-Since it is somehow difficult to be sure about the separation of short-term changes due to winter storms with respect to medium-term changes due to medium-term wave climate, in this filed even a relatively quick reference/analysis of wind and wave data resulting from climate models or satellite-verified database may be of direct help. Although the wave data presented here seems to be sufficient for the proposed medium-term analysis, and a more careful analysis of available wave-data also from global reanalysis or modeling efforts would improve the soundness of analysis but being too heavy, the authors may refer to existing efforts that are represented by available regional cliamte models, originated possibly from efforts such as the MEDATLAS project.*

**We thank the Reviewer for their comment. Though we believe the proposed statistic analysis is appropriate for the scope of our manuscript, i.e. coupling medium-term wave climate and morphological changes, a different approach, like wave-data reanalysis, would be certainly useful but too heavy. However, such aspect will be properly discussed in the text.**

Lorenzoni C., Russo A., Corvaro S., Mancinelli A., & Soldini L. (2017). Comparison between the wintertime and summertime dynamics of the Misa River estuary, Marine Geology, 385, 27-40.

---

## Author Comment (AC3) · 1 Mar 2017

**We appreciate Reviewer $\#2$' comments, which have been taken into due account to improve the clarity and quality of the manuscript. The comments from the Reviewer below are in italic font and our point-by-point responses are in bold.**

*General Comments*
*Summary: This submission investigates medium-term morphodynamics of a barred beach along the Middle Adriatic coast of Italy using annual bathymetric surveys and offshore wave buoy data. Previous studies of similar beaches have focused on short and long-term dynamics, leaving medium-term behavior relatively unstudied. A better understanding of the connections between wave climate and changes in nearshore*

[Figure]

*morphology is needed to improve models that predict storm effects (flooding), beach erosion, or how shoreline protection structures will affect a beach. The authors utilize well-tested data sources (bathymetric surveys and wave buoy from Italian wave measurement network) to examine dynamics on a neglected timescale, medium-term. However, the bathymetric data was collected only once per experimental year, and in my opinion, this limited data set imposes restrictions on the authors' interpretation of the results. Namely, there is insufficient data to separate the possible effects of short term changes due to winter storms from medium-term changes due to medium-term wave climate. See Specific Comments section for further comment.*

**We agree with the Reviewer that only some surveys are available in the period of operation of the wave buoy. However, it should be noticed that the medium-term climate is not a mean climate, but the sum of storms and calm states occurring during the considered time period. Each of these states does give a specific contribution to the overall morphological variation observed in such a period. Hence, we do not want to separate the short-term from the medium-term effects, rather we want to discuss the cumulative effects of all events occurring in that time range. In addition, such an analysis is useful to demonstrate that the beach evolution can be predicted also when a limited number of surveys is available, which is typical for coastal municipalities.**

**Further, it is worth noting that the choice to analyze the medium-term response depends on a number of reasons, mainly: i) when analyzing two consecutive surveys, the distinction between the morphological effects induced by a storm and those induced by calm states is difficult, ii) (short-lasting) winter storms play an important role on the beach dynamics, but this can also be said for typical (long-lasting) summer conditions and infragravity waves, iii) the Adriatic Sea is a long and semi-enclosed basin, characterized by waves coming from multiple directions, even during a specific storm.**

**All of the above points suggest to account for a relatively long time range for a**

**proper estimate of the bar dynamics. Further details are provided in the following responses.**

*Abstract and Title: Abstract explains context and main findings clearly and concisely. The title is appropriate, although upon first reading, the phrase "medium-term dynamics" was unfamiliar. After reading the abstract, I learned that medium-term is a timescale that is longer than short-term (days) and shorter than long-term (years/decades), on the order of seasons or annual. If there is a way to be more clear in the title that medium-term is a timescale, that would be encouraged, but this change is not essential.*

**We prefer to keep the title as it is, as many literature works include the words "medium term" in their titles (e.g., De Vriend et al., 1993[1]; Kuriyama 2002[2]).**

*Organization: This paper is generally well organized. One recommendation- Explain Dean-type beach stability analysis earlier. It is not addressed until the discussion, but it is first mentioned in the "Description of the Site" and then used to explain longshore variability in the "Results". It would be helpful to present the equation with its explanation earlier, so the reader knows where this stable beach shape characterization comes from and why it is relevant for understanding other changes to the beach morphology.*

**We agree with the Reviewer on this point. Hence, we will introduce equation (4) and a brief discussion on the equilibrium profile in Sect.2 ("Description of the Site").**
* * *
[1]De Vriend, H. J., Zyserman, J., Nicholson, J., Roelvink, J. A., Pechon, P., & Southgate, H. N. (1993). Medium-term 2DH coastal area modelling. Coastal Engineering, 21(1-3), 193-224.

[2]Kuriyama, Y. (2002). Medium-term bar behavior and associated sediment transport at Hasaki, Japan. Journal of Geophysical Research: Oceans, 107(C9).

*Specific Scientific Comments*
*Assessment of Medium-term dynamics: Wave climate and nearshore morphology are strongly linked and it is valuable to reveal this relationship over various time scales and environments. The authors focus on a sandy barred beach, chosen for its similarity to many other beaches worldwide, over medium-term time scales. The wave data presented here is sufficient for medium-term analysis. However, a description of the original form of the wave buoy data (time series, hourly product, wave spectra?) and the methods used to further process this data should be specified for the sake of reproducibility.*

**For the sake of clarity, an example of the collected wave data will be illustrated and discussed. Further, the used approach will be better described.**

*The authors acknowledge that beach morphology is "dynamic throughout the year, especially during sea storms driven by NNE winds" (p. 3, line 30). Given this variability, it is necessary to be able to distinguish short-term variability from medium-term variability in order to assess the connection between medium-term wave climate and beach morphology. The bathymetric surveys were collected in different months/seasons each year: June/Summer 2006, February/ Winter 2010, February/Winter 2011, April/Spring 2012, March/Spring 2013. The authors explain that the two types of winds (ESE and NNE) can happen during the same season, and that in the study region, winters are stormy and summers are calm. The winter surveys would therefore be more susceptible to short-term variability due to storms, which is not distinguishable given the limited bathymetric data set. The summer bathymetric surveys are perhaps more representative of medium-term dynamics, because the beach is not subjected to the magnitude or frequency of short-term, high impact events during that season. Since the literature shows and the authors admit that significant bathymetric changes can occur over the course of a single storm, and there is no information given to put each survey in this short-term context, it is not correct to assume that the "snapshot" of bathymetry seen*

*in the data is representative of the medium-term dynamics. The authors should pursue supplementary data (perhaps short-term wave data analysis) to provide context for the bathymetry surveys used in this study.*

**We partially agree with the Reviewer on this point. We know that the bar dynamics strictly depend on short-term events, i.e. storms, as already described in important literature works[3,4], and observed in the studied site[5]. However, the bar dynamics also depend on long-lasting calm states, these occurring in both summer and winter[5]. Since the bar behavior under stormy conditions has already been analyzed in depth, based on either field or laboratory experiments, we here describe the cumulative effects of a series of sea states, i.e. both severe and calm states occurring throughout a significantly long period (about one year), with the aim to illustrate a more comprehensive bar dynamics. It is worth noting that the surveys collected in February 2011 and May 2013 are similar, while surveys collected in February 2011 and February 2010 are significantly different (e.g., see Fig.2), this further demonstrating the independence of the bathymetry on a specific month, and confirming its dependence on the cumulative effects of the wave climate between two consecutive surveys. Finally, preliminary results on the morphological response of the beach of Senigallia subject to a winter storm have already been presented[5,6,7] and a more detailed analysis will be pre-**

[3]Ruessink, B.G., Houwman, K.T., & Hoekstra, P. (1998). The systematic contribution of transporting mechanisms to the cross-shore sediment transport in water depths of 3 to 9 m. Marine Geology, 152(4), 295-324.

[4]Ruessink, B.G., & Terwindt, J.H.J. (2000). The behaviour of nearshore bars on the time scale of years: a conceptual model. Marine Geology, 163(1), 289-302.

[5]Brocchini M., Calantoni J., Postacchini M., Sheremet A., Staples T., Smith J., Reed A.H., Braithwaite III E.F., Lorenzoni C., Russo A., Corvaro S., Mancinelli A., & Soldini L. (2017). Comparison between the wintertime and summertime dynamics of the Misa River estuary, Marine Geology, 385, 27-40.

[6]Calantoni J., Sheremet A., Brocchini M., Postacchini M. (2016). EsCoSed: observations of morphodynamics during Bora at the mouth of the Misa River, 9th International Conference on Multiphase Flow (ICMF). http://www.aidic.it/icmf2016/webpapers/

[7]Palmsten M.L., Calantoni, J., Brocchini M., Soldini L. & Postacchini M. (2016). Sand bar behavior in a mixed

sented in a dedicated work in the near future.

**All of the above will be better clarified in the text and, for the sake of clarity, the temporal evolution of the wave characteristics between two consecutive surveys will be illustrated and analyzed.**

*The authors present current theory, based on peer-reviewed and published field and laboratory measurements, for predicting bar migration based on wave conditions. This theory states that steeper, larger waves promote a seaward shift of the bar and less steep, smaller waves promote a shoreward shift of the bar. The bar migration pattern results presented in this study agree with previous findings. However, agreement with the theory is limited due to insufficient bathymetric data to definitively ascribe morphological changes over a particular year to medium-term wave climate alone (i.e., lacking evidence that short-term wave climate is not contaminating the bathymetric surveys).*

**We partially agree with the Reviewer. We know that only few surveys are available, but the medium-term bathymetric changes are ascribed to the medium-term climate, which includes both short-term events (like storms) and longer states (like calm conditions). Hence, the medium-term changes derive from the cumulative effects of both severe and calm states occurring during the considered temporal range. These considerations will be implemented in the manuscript.**

*Tables: Table 1 & 2: The authors claim it is best to use wave statistics based on the maximum percentage of energy flux over the time interval of interest. This is a fair decision. However, based on the Figure 3 wave roses and Table 1 statistics, there is not always a single band where the energy flux is concentrated. For 2010-2011 especially, the energy flux seems evenly split between $H_{m0} = 1.5 - 2.0m$ (energy flux distribution % of 16.56) and $H_{m0} = 3.0 - 3.5m$ (energy flux distribution % of 16.02).*

sediment environment, Ocean Sciences Meeting. https://agu.confex.com/agu/os16/meetingapp.cgi/Paper/89790

*The authors decide that the dominant waves were about $H_{m0} = 1.75m$. Looking at the wave rose for 2010-2011 there are strong peaks in both the ESE ($25\%$) and NNE ($20\%$) directions. Yet, when summarizing the 2010-2011 period, the authors choose ESE for this time interval. The 2011-2012 and 2012-2013 conditions were truly dominated by one type of wind event over the other, so the assumptions made by the authors for those time intervals are justified. Perhaps a more nuanced bimodal analysis of 2010-2011 is warranted, especially if these bulk wave climate characteristics are used to explain changes in beach morphology.*

**We thank the Reviewer for this comment, which suggests us to clarify the procedure used for the statistic analysis. In particular, it is worth noting that the choice of the ESE forcing is justified by its predominance on the other wave directions, though ESE slightly dominates on NNE forcing. In brief, if we roughly hypothesize that, in 2010-2011, ESE and NNE waves provided the same energy per unit time, the bars would have migrated onshore (due to ESE forcing) for a reasonable time, and offshore (due to NNE forcing) for a slightly smaller time. The balance is an onshore bar migration.**

**The choice of $H_{m0}$ and $T_m$ derives from the below-described procedure:**

- **first, the wave climate during the whole time range is analyzed (i.e., including both storms and calm states) to obtain the energy distribution of Fig.3,**

- **then, the most energetic direction is chosen (in our case the choice is always restricted to ESE or NNE),**

- **finally, the chosen direction (i.e., the 105–135$^o$ sector for ESE or the 15–45$^o$ sector for NNE) is analyzed to obtain the most probable wave characteristics ($H_{m0}$ and $T_m$, respectively shown in Tabs.1 and 2).**

**The above considerations will be implemented in the manuscript.**
*Figures: Figure 6a shows normalized bar height versus normalized bar width with fits for outer and inner bar (essentially steepness curves ($H_{bar}/W_{bar}$) showing how bar geometry changes from outer to intermediate bars). The fits are presented for 2010 and 2013, but not for the other three years of data, leaving the reader questioning whether these trends are consistent.*

**The fitting lines have only been plotted for two years because the other data provide weak best-fit curves. This point will be discussed in the new version of the paper.**

*Furthermore, if the goal is to show that medium-term bar dynamics are strongly linked to medium-term wave climate, it is important to present plots that relate bar features (or changes in bar features) to wave climate metrics (like Table 3, but is visual plot form).*

**We thank the Reviewer for their precious comment. We will try to plot bar feature changes against the wave climate.**

*Figure 6b shows that the cross-shore area of the bar increases southward. A shift in the grain size distribution is the explanation given for the alongshore trend in the equilibrium beach profile. Since grain size distribution is a consideration throughout the authors' analysis of the results, plotting cross-shore bar area versus some grain size distribution metric would be more useful.*

**We partially agree with the Reviewer. In fact, direct measures of sediment size are few and old if compared to the available surveys[8]. However, we will try to produce a graphical output to investigate the sediment-size behavior using the indirect measure obtained from the equilibrium Dean profile.**
* * *
[8]Lorenzoni, C., Mancinelli, A., and Soldini, L.: Caratteristiche sedimentologiche del litorale a Nord di Ancona. Analisi del movimento delle ghiaie (in Italian), in: Atti dell'Istituto di Idraulica dell'Università di Ancona, Università di Ancona, Ancona, p. 54, 1998a.

*Technical Comments*
*Fluency: Although it is apparent that English is the second language of the authors, this does not inhibit the reader's ability to understand this research and its conclusions. There are only a few places where grammar issues impede the authors' message. Listed below are the sentences where a second pass at phrasing would be beneficial. p. 7, line 10 - p. 8, line 3 p. 16, lines 1-7 p. 18, lines 15-20*

**These points will be amended.**

*Equations: Mathematical formulae, symbols, abbreviations, and units are correctly defined and used. References: The number and quality of references is appropriate.*

**We thank the Reviewer for their approval.**

---

## Author Response (AR1)

**Cover Letter**

**Dear Dr. Delhez,**

we have revised the manuscript "Medium-term dynamics of a Middle Adriatic barred beach" by Postacchini et al., which is under consideration for publication in *Ocean Science*.

The revised manuscript addresses all of the comments and suggestions from the three Reviewers, as reported in the following. The highlights of the revisions include: the addition of a new figure, where the volume change between consecutive surveys is illustrated (Fig.6), and a better description of the sediment transport and morphodynamics, as required by Reviewer #1; a better description of the medium-term approach, an example of the examined time series (new panel of Fig.3), the illustration of the shape parameter A (Fig.5b) and the estimate of the bar-geometry change (Tab.3), as required by Reviewer #2; additional references suggested by Reviewer #3.

Further, we took advantage of this revision to amend some typos and significantly improve the clarity and quality of the manuscript.

We hope you will find the revised manuscript suitable for publication. We look forward to your response.

Yours sincerely, Matteo Postacchini

**Response to Reviewers**

**Reviewer #1**

We thank Reviewer #1 for their useful comments and suggestions which will help to improve our manuscript. The comments from the Reviewer below are in italic font and our point-by-point responses are in bold.

**General comments**

The paper focuses on the morphodynamic analysis of a natural sandy littoral stretch located along the highly urbanized coastal sector of the Western Adriatic Sea. Morphologic variability of submerged sandbars is quantitatively analysed and compared with in situ wave data to evaluate the near-shore mediumterm dynamics. The study domain is located in a semi-enclosed and elongated basin whose coastal dynamics are deeply influenced by the physiographical setting. The paper furnishes new insight on local (Central Adriatic) and general (semi-enclosed basin) bars behaviour in medium-term. For this reason, and with the aim to extend the local findings to more general ones, the authors use proper conceptual and analytical methods to describe data and results. However, in order to fully constrain the reached conclusions, a better description of some evidences on the sandbars morphologic variability is needed.

We thank the Reviewer for their appreciation of our analysis. As suggested below, the new version of the manuscript does better describe the evidences of sandbar migration and morphological variability of the beach.

Specific comments

**"Sandbar cross-shore migration evidences"**

In the section 4.2 "bathymetric surveys", the proposed seaward/shoreward sandbars migration patterns between consecutive surveys (2006, 2010, 2011, 2012, 2013) are not clearly detectable. Considering the importance of this phenomenon in relation to the paper objective, a better description and presentation are required. Referring to Figure 5, some considerations are reported below. The 2006-2010 shoreward migration is highly localized around the "rotonda sections", the southernmost transects on the contrary seem not to highlight any kind of real net cross-shore displacement. The 2010-2011 evolution is represented by a detectable sandbars cross-shore displacement but not uniform alongshore. The Authors propose a shoreward migration, thus assuming that the sand volume stored in 2010 on the middle-bar had contributed to the upper profile evolution and the large sand volume stored in 2011 on the outer-bar would be alongshore derived (see for example Figure 2). On the contrary, a seaward displacement would imply that the sand volume stored in 2010 on the middle-bar had contributed to the lower profile evolution and the net alongshore sand movement would be located in shallower depth. Even if this is a very limited example, qualitatively detected by a single transect, a volumetric approach could be fundamental to evaluate the

**"cross-transect and along-transect" sedimentary mobility, thus improving the alongshore consistency of interpreted cross-shore sandbars displacement.**

We thank the Reviewer for their suggestions, which helped us to better highlight the objective of our paper and also to find some errors in the original version of the paper. In particular, the bar data of Fig.5 have been updated and are now consistent with Fig.4 (in the original version, preliminary data were used). Further, Fig.4 has been changed to make it clearer, while the description of the sandbar migration has been improved (see Sect. 4.2). In addition, a new figure (Fig.6 of the present manuscript) has been added, to clearly show the differences in the beach evolution between 2010 and 2011: representative cross-shore profiles have been illustrated (near and far from the rigid structures), together with the volume-change evolution. Far from the structures, where it seems that all bars migrate shoreward with the inner bar feeding the upper beach, the alongshore sediment transport seems to be negligible, while it is important nearby the "Rotonda". A detailed description has been included at the end of Sect. 4.2.

**"Sandbar alongshore variability"**

As stated by the Authors, since the area is located close to the jetty and characterized by complex hydrodynamics, it could be likely to develop complex sandbar morphologies. Moreover during particular storm events, the coupling of complex hydrodynamics and morphologies could induce cross-shore beach profile response, not necessary in phase with areas farther away from "jetty-rotonda" (see for example Shand et alii, 2001), thus the generalization on the bars behaviour along the whole area could be more complex to define. In this framework, as noted by the Authors, the stabilization of the "bar features" along the southernmost transects could testify the boundary between the near-shore area influenced and "not directly influenced" by the jetty and the "rotonda" structures. The same applies to the cross-shore limits of "rotonda" influence on sandbar characteristics, as well evidenced and described in Figure 5b.

The three-dimensional bar behavior has been better explained, also with reference to the influence of the rigid structures. In addition, cross-shore profiles and volume change helped in the overall interpretation of the alongshore sediment transport. A detailed description has been included at the end of Sect. 4.2.

**Reviewer #2**

We appreciate Reviewer #2' comments, which have been taken into due account to improve the clarity and quality of the manuscript. The comments from the Reviewer below are in italic font and our point-by-point responses are in bold.

**General Comments**

Summary: This submission investigates medium-term morphodynamics of a barred beach along the Middle Adriatic coast of Italy using annual bathymetric surveys and offshore wave buoy data. Previous studies of similar beaches have focused on short and long-term dynamics, leaving medium-term behavior relatively unstudied. A better understanding of the connections between wave climate and changes in nearshore morphology is needed to improve models that predict storm effects (flooding), beach erosion, or how shoreline protection structures will affect a beach. The authors utilize well-tested data sources (bathymetric surveys and wave buoy from Italian wave measurement network) to examine dynamics on a neglected timescale, medium-term. However, the bathymetric data was collected only once per experimental year, and in my opinion, this limited data set imposes restrictions on the authors' interpretation of the results. Namely, there is insufficient data to separate the possible effects of short term changes due to winter storms from medium-term changes due to medium-term wave climate. See Specific Comments section for further comment.

We agree with the Reviewer that only some surveys are available in the period of operation of the wave buoy. However, it should be noticed that the medium-term climate is the sum of storms and calm states occurring during a specific time period. Each state provides a specific contribution to the overall morphological variation observed in such a period. Hence, the short-term events cannot be separated from the medium-term climate, rather we want to discuss the cumulative effects of all events occurring in a specific time range. Such an analysis is also useful to demonstrate that the beach evolution can be fairly well predicted when a limited number of surveys is available, which is typical for coastal municipalities.

Further, it is worth noting that the choice to analyze the mediumterm response depends on a number of reasons, mainly: i) when analyzing two consecutive surveys, the distinction between the morphological effects induced by a storm and those induced by calm states is difficult, ii) the Adriatic Sea is a long and semi-enclosed basin, which is characterized by an extremely variable climate, with significantly large deviations of the wave characteristics from the mean values, even during a single storm.

All of the above suggest to account for a relatively long time period for a proper estimate of the bar dynamics in the western Adriatic. These points are now clarified in the text and further details are provided in the following responses. Abstract and Title: Abstract explains context and main findings clearly and concisely. The title is appropriate, although upon first reading, the phrase "medium-term dynamics" was unfamiliar. After reading the abstract, I learned that medium-term is a timescale that is longer than short-term (days) and shorter than long-term (years/decades), on the order of seasons or annual. If there is a way to be more clear in the title that medium-term is a timescale, that would be encouraged, but this change is not essential.

We prefer to keep the title as it is, as many literature works include the words "medium term" in their titles (e.g., De Vriend et al.,  $1993^{1}$ ; Kuriyama  $2002^{2}$ ).

Organization: This paper is generally well organized. One recommendation-Explain Dean-type beach stability analysis earlier. It is not addressed until the discussion, but it is first mentioned in the "Description of the Site" and then used to explain longshore variability in the "Results". It would be helpful to present the equation with its explanation earlier, so the reader knows where this stable beach shape characterization comes from and why it is relevant for understanding other changes to the beach morphology.

We agree with the Reviewer on this point. Hence, we have introduced the discussion on the equilibrium beach profile and the related equation in Sect. 2 ("Description of the Site").

**Specific Scientific Comments**

Assessment of Medium-term dynamics: Wave climate and nearshore morphology are strongly linked and it is valuable to reveal this relationship over various time scales and environments. The authors focus on a sandy barred beach, chosen for its similarity to many other beaches worldwide, over medium-term time scales. The wave data presented here is sufficient for medium-term analysis. However, a description of the original form of the wave buoy data (time series, hourly product, wave spectra?) and the methods used to further process this data should be specified for the sake of reproducibility.

For the sake of clarity, an example of the significant height obtained from the waverider is illustrated in Fig.3e and discussed in Sect. 4.1. The used approach has been better described in the same section.

The authors acknowledge that beach morphology is "dynamic throughout the year, especially during sea storms driven by NNE winds" (p. 3, line 30). Given this variability, it is necessary to be able to distinguish short-term variability from medium-term variability in order to assess the connection between medium-term wave climate and beach morphology. The bathymetric surveys were collected in different months/seasons each year: June/Summer 2006, February/

<sup>1De Vriend, H. J., Zyserman, J., Nicholson, J., Roelvink, J. A., Pechon, P., & Southgate, H.

N. (1993). Medium-term 2DH coastal area modelling. Coastal Engineering, 21(1-3), 193-224. 2Kuriyama, Y. (2002). Medium-term bar behavior and associated sediment transport at Hasaki, Japan. Journal of Geophysical Research: Oceans, 107(C9).

Winter 2010, February/Winter 2011, April/Spring 2012, March/Spring 2013. The authors explain that the two types of winds (ESE and NNE) can happen during the same season, and that in the study region, winters are stormy and summers are calm. The winter surveys would therefore be more susceptible to short-term variability due to storms, which is not distinguishable given the limited bathymetric data set. The summer bathymetric surveys are perhaps more representative of medium-term dynamics, because the beach is not subjected to the magnitude or frequency of short-term, high impact events during that season. Since the literature shows and the authors admit that significant bathymetric changes can occur over the course of a single storm, and there is no information given to put each survey in this short-term context, it is not correct to assume that the "snapshot" of bathymetry seen in the data is representative of the medium-term dynamics. The authors should pursue supplementary data (perhaps short-term wave data analysis) to provide context for the bathymetry surveys used in this study.

We partially agree with the Reviewer on this point and what follows has been clarified in Sect. 3. The bar dynamics strictly depend on short-term events, as already observed worldwide3, but also on longlasting calm states occurring in both summer and winter4. Hence, we here describe the cumulative effects of a series of events, i.e. both energetic and calm states occurring during a significantly long period (about one year), with the aim to illustrate a more comprehensive bar dynamics. Further, it is worth noting that the surveys collected in February 2011 and May 2013 are similar, while surveys collected in February 2011 and February 2010 are significantly different (e.g., see Fig.2), this demonstrating the independence of the bathymetry on a specific month/season, and confirming its dependence on the cumulative effects of the wave climate between two consecutive surveys.

Finally, preliminary results on the morphological response of the beach of Senigallia subject to a winter storm have already been presented4,5,6 and a more detailed analysis will be illustrated in a dedicated work in the near future.

The authors present current theory, based on peer-reviewed and published

<sup>3Ruessink, B.G., Houwman, K.T., & Hoekstra, P. (1998). The systematic contribution of transporting mechanisms to the cross-shore sediment transport in water depths of 3 to 9 m. Marine Geology, 152(4), 295-324.

<sup>4Brocchini M., Calantoni J., Postacchini M., Sheremet A., Staples T., Smith J., Reed A.H., Braithwaite III E.F., Lorenzoni C., Russo A., Corvaro S., Mancinelli A., & Soldini L. (2017). Comparison between the wintertime and summertime dynamics of the Misa River estuary, Marine Geology, 385, 27-40.

<sup>5Calantoni J., Sheremet A., Brocchini M., Postacchini M. (2016). EsCoSed: observations of morphodynamics during Bora at the mouth of the Misa River, 9th International Conference on Multiphase Flow (ICMF). http://www.aidic.it/icmf2016/webpapers/

<sup>6Palmsten M.L., Calantoni, J., Brocchini M., Soldini L. & Postacchini M. (2016). Sand bar behavior in a mixed sediment environment, Ocean Sciences Meeting. https://agu.confex.com/agu/os16/meetingapp.cgi/Paper/89790

field and laboratory measurements, for predicting bar migration based on wave conditions. This theory states that steeper, larger waves promote a seaward shift of the bar and less steep, smaller waves promote a shoreward shift of the bar. The bar migration pattern results presented in this study agree with previous findings. However, agreement with the theory is limited due to insufficient bathymetric data to definitively ascribe morphological changes over a particular year to medium-term wave climate alone (i.e., lacking evidence that short-term wave climate is not contaminating the bathymetric surveys).

We partially agree with the Reviewer. We know that only few surveys are available, but the medium-term bathymetric changes are ascribed to the medium-term climate, which includes both shortterm events (like storms) and longer states (like calm conditions). Hence, the medium-term changes derive from the cumulative effects of both severe and calm states occurring during the considered temporal range. These considerations have been implemented in Sect. 3.

Tables: Table 1 & 2: The authors claim it is best to use wave statistics based on the maximum percentage of energy flux over the time interval of interest. This is a fair decision. However, based on the Figure 3 wave roses and Table 1 statistics, there is not always a single band where the energy flux is concentrated. For 2010-2011 especially, the energy flux seems evenly split between  $H_{m0} = 1.5 -$ 2.0m (energy flux distribution % of 16.56) and  $H_{m0} = 3.0 - 3.5m$  (energy flux distribution % of 16.02). The authors decide that the dominant waves were about  $H_{m0} = 1.75m$ . Looking at the wave rose for 2010-2011 there are strong peaks in both the ESE (25%) and NNE (20%) directions. Yet, when summarizing the 2010-2011 period, the authors choose ESE for this time interval. The 2011-2012 and 2012-2013 conditions were truly dominated by one type of wind event over the other, so the assumptions made by the authors for those time intervals are justified. Perhaps a more nuanced bimodal analysis of 2010- 2011 is warranted, especially if these bulk wave climate characteristics are used to explain changes in beach morphology.

We thank the Reviewer for this comment, which suggests us to clarify the procedure used for the statistic analysis, now better described in Sect. 4.1. The choice of the ESE forcing is justified by its predominance on the other wave directions, though ESE only slightly dominates on NNE forcing. A brief analysis of the NNE direction is also undertaken for 2010-2011, this confirming that the NNE sector generally provides steeper waves if compared to those characterizing the ESE sector, whether or not this represents the most energetic sector. In addition, the splitting of the energy flux is now better discussed in Sect.4.1 (please note that Tab.1 and Tab. 2 refer to to 2012-2013, and not to 2010-2011, as incorrectly stated in the original version). In particular, the choice of the lower wave-height range is motivated by a larger wave frequency characterizing that class. Figures: Figure 6a shows normalized bar height versus normalized bar width with fits for outer and inner bar (essentially steepness curves  $(H_{bar}/W_{bar})$  showing how bar geometry changes from outer to intermediate bars). The fits are presented for 2010 and 2013, but not for the other three years of data, leaving the reader questioning whether these trends are consistent.

The fitting lines have only been plotted for two years as the other data provide weak best-fit curves. This point is discussed in the new version of the paper.

Furthermore, if the goal is to show that medium-term bar dynamics are strongly linked to medium-term wave climate, it is important to present plots that relate bar features (or changes in bar features) to wave climate metrics (like Table 3, but is visual plot form).

We thank the Reviewer for their comment. We have preferred to include the outer bar changes in Tab.3, where they are easily comparable with the wave climate (notice that the wave propagation was performed from the offshore to the outer bar depth). Brief discussions have been included at the end of Sect.4.4 and 5.

Figure 6b shows that the cross-shore area of the bar increases southward. A shift in the grain size distribution is the explanation given for the alongshore trend in the equilibrium beach profile. Since grain size distribution is a consideration throughout the authors' analysis of the results, plotting cross-shore bar area versus some grain size distribution metric would be more useful.

As also stated in the manuscript, direct measures of sediment size are very few and older if compared to the available surveys. Further, we have tried to plot the cross-shore bar area against the A parameter, which is an indirect measure of the sediment size, but this gives a weak correlation. However, the alongshore evolution of A has been included in Fig.7b, using a secondary axis. This is properly discussed in Sect. 4.3 and 5.

**Technical Comments**

Fluency: Although it is apparent that English is the second language of the authors, this does not inhibit the reader's ability to understand this research and its conclusions. There are only a few places where grammar issues impede the authors' message. Listed below are the sentences where a second pass at phrasing would be beneficial. p. 7, line 10 - p. 8, line 3 p. 16, lines 1-7 p. 18, lines 15-20

These points have been amended.

Equations: Mathematical formulae, symbols, abbreviations, and units are correctly defined and used. References: The number and quality of references is appropriate.

We thank the Reviewer for their approval.

**Reviewer #3**

We thank Reviewer #3 for their precious suggestions, which will be implemented to improve the quality of our manuscript. The comments from the Reviewer below are in italic font and our point-bypoint responses are in bold.

I have read with interest the paper "Medium-term dynamics of a middle Adriatic barred beach" by Postacchini et al. The MS paper deals with the morphodynamic analysis of a natural sandy littoral stretch located along a highly urbanized coastal sector on the west Adriatic Sea. The morphologic variability of submerged sandbars is analysed and compared with in situ wave data, on order to evaluate medium-term dynamics. Generally speaking, I find the paper interesting, tackling an important aspect in a convincing way (although of course dealing with the usual problems of having "not enough" data) and well organized. The English is also rather fluent.

We thank again the Reviewer for their appreciation of our work.

However, the MS may benefit from some minor improvements that could be easily implemented by the authors.

- I feel the need of more specific links to some of the several existing waveclimate studies on the Adriatic basin. - The paper tackles a subject relatively unexplored, i.e. the medium-term behavior, that is a timescale that between short term (days) and long term (years/decades), on the order of seasons or years, using annual bathymetric surveys and offshore wave buoy data. There is no doubts that improving the knowledge on the relations between wave climate and changes in nearshore morphology is a necessary step, in order to improve numerical models capable of predicting storm effects, beach erosion, and more generally the efficiency of shoreline protection measures. This is even more valid in a context of climate change, that should be also mentioned in the paper more clearly. Benetazzo et al. 2012, DOI:10.5194/nhess-12-1-2012, and references therein inlcluded may give some useful hints on this.

We agree with the Reviewer. What suggested has been included in the first paragraph of Sect.1.

- at the same time, some lines addressing the realtionship between the local scale dynamics with a more regional scenario of sediment dynamics and transport should possibly be introduced by the authors. Sherwood et al., 2004, Oceanography; or Harris et al., DOI: 10.1029/2006JC003868. Even though not strictly pertinent to the study, it should be indeed mentioned that the longshore and cross-shore budget of the local beach is however to be framed within a more regional dynamics.

To better contextualize the observed morphological changes in the Adriatic framework, we have introduced the suggested regional aspects at the end of Section 2. - other existing approaches could be mentioned in order to provide a more complete series of wave data nearshore, including the transfer of offshore wave data to the coast by means of wave models, e.g. SWAN, in order to reconstruct a more detailed and spatially meaningful wave climate (Carniel et al., 2011, DOI: 10.2478/s13545-011-0036-1)

This point has been amended as suggested.

- Some more caveats should be discussed, since the bathymetric data was collected once per year and are therefeore somehow limited. Possible workarounds could be put more in evidence, as the use of video images to reconstruct the coastline (see Archetti et al., 2016, . doi:10.5194/nhess-16-1107-2016, and references therein included.

The suggested references have been included in the Conclusions, where the coastal video monitoring is discussed as a possible improvement of the present analysis.

- Since it is somehow difficult to be sure about the separation of shortterm changes due to winter storms with respect to medium-term changes due to medium-term wave climate, in this filed even a relatively quick reference/analysis of wind and wave data resulting from climate models or satellite-verified database may be of direct help. Although the wave data presented here seems to be sufficient for the proposed medium-term analysis, and a more careful analysis of available wave-data also from global reanalysis or modeling efforts would improve the soundness of analysis but being too heavy, the authors may refer to existing efforts that are represented by available regional cliamte models, originated possibly from efforts such as the MEDATLAS project.

We thank the Reviewer for their suggestion, which has been included in the Conclusions.

**Relevant changes**

- New figure for the comparison of the cross-shore profiles and the estimate of the volume change through a beach profile.
- Inclusion of a wave-height time series recorded by the waverider.
- Illustration of the shape parameter evolution.
- Estimate of bar alongshore-averaged geometry changes.
- Better description of the beach morphodynamics and medium term approach.

[revised manuscript text omitted]

---

## Referee Report (RR1)

**General comments:** The paper seeks to analyze the medium-term (seasons to years) morphological evolution of sandy multiple-barred beaches using bathymetry observations at Senigallia in 2006 and annually from 2010 to 2013, and nearby wave buoy observations from 2010. The need for this analysis is well motivated and the kinematic/geometric description of the bar behavior is interesting. However, a more direct quantitative link needs to be made between the wave climate and morphological change to support the central conclusion. While the writing is good overall, the paper would be improved by text edits for grammar and clarity.

**Specific scientific comments:**

"Medium-term dynamics": This term is defined nicely in the conclusions section. It seems important to introduce this early in the paper, both in the abstract and introduction, to provide a concrete definition for timescales and types of features that are the focus of the paper.

"Cumulative effects", overview: The hypothesis that the medium term response can be considered a result of the sum of the contributions of all wave events in some time range is interesting and worthy of study. My understanding is that an additional related hypothesis considered by the authors is that the contributions of all wave events in some time range can be parameterized by a single representative wave condition, which they define in Section 4.1. Specifically, they hypothesize that medium-term wave conditions that are characterized by steeper and larger waves are correlated with seaward bar migration, while medium-term wave conditions that are less steep and smaller are correlated with shoreward bar migration. This relationship has been shown for short-term bar response considering the short-term wave statistic. As the authors point out, if this relationship holds using medium-term wave statistics, this could be a powerful tool to predict bar changes from one time to the next without directly considering shorter-term changes in between.

"Cumulative effects", point 1: While the three data points provided (2010-2011, 2011-2012, and 2012-2013) are roughly qualitatively consistent with this relationship (medium-term bar response and medium-term wave statistic), a more quantitative test should be performed and the discussion should be clarified. The authors could attempt to make a more direct quantitative comparison by (1) computing a single representative (alongshore-averaged) bar position for each year, (2) subtracting those positions to estimate the change, and (3) plotting that change as a function of the wave steepness. Similar analysis already has been done for the outer bar geometry in Table 3 (this may be clearer if shown graphically). Even if that approach is pursued, it likely will be difficult to establish correlation with only three data points. Additional data could be sought by (1) adding the 2006 to 2010 time window by obtaining nearby wave observations or model hindcasts or (2) extending the analysis forward to 2017 using new surveys and/or video observations in 2015-2017.

"Cumulative effects", point 2: It is possible that details of the wave climate in between intervening medium-term time periods that are not captured by the authors' single metric may be important to the bar states observed at various times in 2006-2013. To test this hypothesis, perhaps the time series of a wave metric related to bar migration could be integrated in time to achieve a wave metric that captures the "cumulative" effect the authors discuss. This could be compared with the simpler bulk estimate the authors describe in Section 4.1. If video observations are included, the hypotheses about the relationship between short-term and medium-term dynamics could be tested more rigorously.

**Specific technical comments:**

While much of the writing and organization are good, the readability could be improved significantly by fixing grammatical errors and confusing phrases throughout the paper. Below I list examples in the introduction. Similar revisions could be made in the other sections.

P1 L18: word "and" is missing at end of list

P1 L19-20: phrase "strictly related to the above-mentioned aspects" could be made clearer

P1 L20: unclear what "also" refers to. Meaning may be both summer and winter?

P1 L22: unclear what "also" refers to. Meaning may be "nearshore dynamics *including* rapid morphological changes to the beach"?

P2 L3: word "due" seems unneeded

P2 L9: "both" should be removed (list of three methods)

P2 L11: "experiences" maybe should be replaced with "experiments"

P2 L17&20 and P3 L8: word "this" not needed

P2 L23: "have" should be replaced with "has"

---

## Author Response (AR2)

**Cover Letter**

Dear Dr. Delhez,

we have revised the manuscript "Medium-term dynamics of a Middle Adriatic barred beach" by Postacchini et al., which is under consideration for publication in *Ocean Science.*

We have tried to follow all Reviewers' suggestions, which helped us to improve the manuscript.

While Reviewers #2 and #5 commented positively our work and required some additions and clarifications, Reviewer #4 suggested us to significantly modify the manuscript by including further elaborations, especially using new morphological and wave data. Although video-observations cannot be included in the present work (elaborations are still underway and will be published in dedicated papers), we have tried to 1) use model hindcasts and 2) search for recent bathymetries. However, validation of model hindcasts with available buoy data led to very weak correlations, while only few bathymetric data have been found, i.e. one cross-shore profile with information up to about 2m depth, which is insufficient for our purposes (see figure in the reply).

However, due to the positive comments in both previous and present (Reviewers #2 and #5) reviews, and due to our significant efforts to improve the paper and satisfy all Reviewers' comments, we believe that the work now merits to be shared within the scientific community.

We look forward to hearing from you in due course.

Yours sincerely,
Matteo Postacchini

**Response to Reviewers**

**Report #1 - Reviewer #4**

General comments: The paper seeks to analyze the medium-term (seasons to years) morphological evolution of sandy multiple-barred beaches using bathymetry observations at Senigallia in 2006 and annually from 2010 to 2013, and nearby wave buoy observations from 2010. The need for this analysis is well motivated and the kinematic/geometric description of the bar behavior is interesting. However, a more direct quantitative link needs to be made between the wave climate and morphological change to support the central conclusion. While the writing is good overall, the paper would be improved by text edits for grammar and clarity.

**We thank the Reviewer for their comments and appreciation for our work. A point-by-point reply is provided in bold, including references to the revised manuscript.**

Specific scientific comments:

"Medium-term dynamics": This term is defined nicely in the conclusions section. It seems important to introduce this early in the paper, both in the abstract and introduction, to provide a concrete definition for timescales and types of features that are the focus of the paper.

**We agree with the Reviewer and introduced the "medium-term" concept in both abstract and final part of the Introduction.**

"Cumulative effects", overview: The hypothesis that the medium term response can be considered a result of the sum of the contributions of all wave events in some time range is interesting and worthy of study. My understanding is that an additional related hypothesis considered by the authors is that the contributions of all wave events in some time range can be parameterized by a single representative wave condition, which they define in Section 4.1. Specifically, they hypothesize that medium-term wave conditions that are characterized by steeper and larger waves are correlated with seaward bar migration, while medium-term wave conditions that are less steep and smaller are correlated with shoreward bar migration. This relationship has been shown for short-term bar response considering the short-term wave conditions, but not for medium-term response considering a medium-term wave statistic. As the authors point out, if this relationship holds using medium-term wave statistics, this could be a powerful tool to predict bar changes from one time to the next without directly considering shorter-term changes in between.

"Cumulative effects", point 1: While the three data points provided (2010-2011, 2011-2012, and 2012-2013) are roughly qualitatively consistent with this relationship (medium-term bar response and medium-term wave statistic), a more quantitative test should be performed and the discussion should be clarified. The authors could attempt to make a more direct quantitative comparison by (1) computing a single representative (alongshore-averaged) bar position for

each year, (2) subtracting those positions to estimate the change, and (3) plotting that change as a function of the wave steepness. Similar analysis already has been done for the outer bar geometry in Table 3 (this may be clearer if shown graphically).

**We partially agree with the Reviewer on this point. We have tried to plot the displacement of the bar-crest location against the wave steepness for both inner, intermediate and outer bars, obtaining the result illustrated in the following figure), which does not suggest further relevant conclusions. However, information on the bar migration, i.e. the displacement of the bar-crest location $\Delta\overline{s_{cr}}$, has been included in the manuscript (see Tab.3 and end of sect.4.4).**

[Figure]

**Wave steepness vs bar migration.**

Even if that approach is pursued, it likely will be difficult to establish correlation with only three data points. Additional data could be sought by (1) adding the 2006 to 2010 time window by obtaining nearby wave observations or model hindcasts or (2) extending the analysis forward to 2017 using new surveys and/or video observations in 2015-2017.

**Since the buoy worked till March 2014 and data between 2006 and 2010 are missing, a first attempt to use modeled data/hindcasts (e.g., ECMWF-type or NOAA-type data) have been made, but with poor results. In particular, the comparison between available buoy data and model hindcast-forecast data is fairly weak, in terms of both time series and wind rose/wave climate referring to multi-year periods. As an example, the model wave frequencies are much more peaked in the main sectors (up to 4-5 times larger) if compared to the buoy data. Hence, the alternative use of buoy and model data is not suitable for**

the analysis of bar migration.

Further, concerning the available surveys, after 2013 only one cross-shore profile was collected by the Marche Region in the analyzed domain. This is located $\sim 90m$ North of the "Rotonda" and is available for 2015 and 2017, spanning from the emerged beach to a depth of $\sim 2m$ (see following figure), providing very poor (and probably misleading) information on the bar dynamics, especially when associated to the present analysis.

In addition, the elaboration of available video observations, which is still underway, will be illustrated in dedicated works in the near future.

Finally, further and appropriate bathymetric surveys are not available, while alternative wave data are not reliable/suitable and video elaborations are still underway, this suggesting us to leave the analysis as it is. We are confident that our work can represent the starting point for future analyses.

[Figure]

**Cross-shore profile located $\sim 90m$ North of the "Rotonda", surveyed in 2015.**

"Cumulative effects", point 2: It is possible that details of the wave climate in between intervening medium-term time periods that are not captured by the authors' single metric may be important to the bar states observed at various times in 2006-2013. To test this hypothesis, perhaps the time series of a wave metric related to bar migration could be integrated in time to achieve a wave metric that captures the "cumulative" effect the authors discuss. This could be compared with the simpler bulk estimate the authors describe in Section 4.1. If video observations are included, the hypotheses about the relationship between short-term and medium-term dynamics could be tested more rigorously.

We thank the Reviewer for their comments, which has helped us to better explain the purpose of our methodology. It is worth noting that use of a procedure like that suggested by the Reviewer (integration in time of "the time series of a wave metric related to

bar migration") may lead to an inaccurate wave descriptor. As an example, the coupling and integration of the available time series (e.g., wave height and period) with the aim to roughly estimate an averaged or integrated quantity (e.g. the wave power), but without any information on the wave direction, which is crucial in our analysis.

On the other hand, our methodology is both simple and complete, as this accounts for all available buoy data (direction, height, period) and elaborates them with the purpose to account for all processes occurred during the considered time intervals, starting from the well-known assumption that significantly different physics generate in the studied area during NNE and ESE waves. This has been better highlighted in Sect.4.1.

Video observations cannot be used for the above-mentioned reasons (see previous reply).

Specific technical comments:

While much of the writing and organization are good, the readability could be improved significantly by fixing grammatical errors and confusing phrases throughout the paper. Below I list examples in the introduction. Similar revisions could be made in the other sections.

P1 L18: word "and" is missing at end of list

P1 L19-20: phrase "strictly related to the above-mentioned aspects" could be made clearer

P1 L20: unclear what "also" refers to. Meaning may be both summer and winter?

P1 L22: unclear what "also" refers to. Meaning may be "nearshore dynamics including rapid morphological changes to the beach"?

P2 L3: word "due" seems unneeded

P2 L9: "both" should be removed (list of three methods)

P2 L11: "experiences" maybe should be replaced with "experiments"

P2 L17 & 20 and P3 L8: word "this" not needed

P2 L23: "have" should be replaced with "has"

The above-listed points have been amended in the introduction and throughout the rest of the paper.

**Report #2 - Reviewer #2**

Summary

This referee acknowledges significant changes to the manuscript and commends the authors for their diligent revisions. The responses below are specific to previous comments from "Anonymous Referee #2". With minor technical changes and minor suggestions for revision, this referee recommends this research paper for publication.

**We thank the Reviewer for their useful comments and appreciation for the revised paper. In the following, a point-by-point reply to their comments is provided in bold.**

General & Specific Comments

The extended explanation of medium-term dynamics, with careful distinction of factors that can be teased apart and those that remain indistinguishable is appreciated. It is now my understanding, thanks to the authors revision, that even though the change in bathymetry between annual surveys could be/sometimes is influenced by a dramatic storm just before the survey, this storm is incorporated into the medium-term dynamics analysis, and that the changes in the annual bathymetry are of a different magnitude than what has been observed from a single storm. Therefore, the conclusions drawn from the differencing of bathymetric maps between years represents the cumulative effect of calm and stormy conditions (varied directional forcing is also accounted for) without being biased by short-term changes. Is my understanding correct? If so, I think one additional comment on the relative magnitude of observed short-term changes to a bar system with respect to medium-term changes would be appropriate. It does not need to come from this field site- observations from a similarly sandy, barred beach would suffice.

**The Reviewer is correct. As suggested, the daily (short-term) and yearly (medium-term) bar migration rates, as well as appropriate references, have been included in Sect.3. Migration rates in the analyzed site are included in the end of sect.4.4.**

The explanation of the Dean-type equilibrium beach profile is presented early and in the context of this field site. The authors' changes clarify that the study beach has been found to match the Dean-type equilibrium profile (shape parameter A is relatively constant in time) and that the profile varies as expected with the changing $d_{50}$.

Thank you for specifying your wave time series and spectral processing. These details substantiate the decisions you made to analyze specific bands in wave height and wave period.

The addition of the outer bar change parameters to Table 3 and the discussion of those parameters in the adjacent text clarify your message about the link between wave and wind climate and the bar morphodynamics.

**We thank the Reviewer for their appreciation for our improvements.**

In section 4.2 Bathymetric surveys, on p. 14, the paragraph beginning with "The influence of the Rotonda": In this section, the authors describe bathymetric variation local to the engineered structures and ascribe these changes in specific years to the presence of the structures. I am assuming this influence was expected due to the predictable effects of engineered structures on sediment migration under given wave conditions, but this link is not explicitly made by the authors. Perhaps it is more appropriate to limit the results section to describing the changes and stating that they were observed local to the structures, and then discuss why this was expected and how the data confirms this prediction in the Discussion section.

**We thank the Reviewer for their useful comment. We improved the manuscript as suggested, changing both Results and Discussion sections.**

Technical Comments

p. 1, line 3: "area in the Region" does not need a capital "R".
**Amended as suggested.**

p. 14, line 26: typo- doubled dash between 25 and 40%.
**Amended as suggested.**

p. 20, line 8: "seward" change to "seaward"
**Amended as suggested.**

p. 20, line 9-11: "rather than by" suggests that the 2012-2013 ESE climate does not produce significant changes, but it does. Perhaps you are trying to say that the bimodal wave climate in 2010-2011 does not produce large changes, but the NNE and ESE climates do produce large magnitude changes in opposite directions. If so, try rewording "rather than by" to "or by".
**Amended as suggested.**

p. 22, line 32: "suggested us to" is not quite correct; use "suggested we" or "led us to".
**Amended as suggested.**

p. 23, line 11: "Lagrangian dirfters, able at measuring" change to "capable of measuring".
**Amended as suggested.**

**Report #3 - Reviewer #5**

Overview:

The paper under review couples the movement of subtidal bars along a stretch of the Adriatic to wind wave forcing with respect to medium-term time scales. The area of study is significant because 1) it is located along the coast of a semi-enclosed basin, 2) has a small tidal range, and 3) a range of wave forcing that deviates greatly from the average. The effects of short- and long-term dynamics appear relatively frequently in literature. In the present study, the authors consider the cumulative effects of middle-term timescales (on the order of years). Using data from an offshore, waverider buoy and bathymetric surveys collected in consecutive years, the author's link the bar movement between surveys to the breaking of wind waves.

**We thank the Reviewer for their comments. A point-by-point reply is provided in bold, including line-number references to the revised manuscript.**

Approach:

1) I understand the authors' argument of cumulative effects when referring to medium-term dynamics and responses. However, I cannot completely convince myself that any given survey is not skewed toward the most recent wave event or lack thereof. In the first sentence of the conclusion, the authors suggest that medium-term evolution falls between decadal and event timescales. As presented, the analyses focus on the periods between available surveys. It would be convincing to see that surveys from 2011 or 2012 can be "predicted" given the other available surveys and the wave data.

**The cumulative effects are fundamental for the analysis at hand and every event is important. For sure, the surveyed bathymetry is significantly influenced by the wave climate characterizing the time period just before the survey, whether it is characterized by a stormy event or by a calm period. However, it is only a part of the whole climate characterizing the considered time range.**

**Further, our method may help to predict the bar migration, but not the complete survey. In the manuscript, we have validated the predictive methodology by comparing the bar migration direction and a characteristic relative wave height.**

2) While the steps of the wave statistical analysis given on page 9 are useful, I would like to know more about how the wave data were processed. It is not clear if the authors used bulk wave parameters or processed spectral data from the buoy.

**The wave data are first processed by the buoy, which provides spectral data (significant height, peak period, etc.) every half an hour. This has been specified in Sect.4.1.**

3) The statistical analyses of the wave data provide estimated values of wave parameters for the directional bins of interest to the authors. Given the statistical procedure, it is not accurate to refer to the values as "mean" values (page 10, lines 13 and 22). It appears that the "mean" values reported were determined by assigning the middle value of the bin chosen by the authors (table 2). Actual mean values can be reported by taking the average of all the values that fall into the bins. If this is how the mean was determined, it should be made clearer in the text. Furthermore, mean values should be denoted in the text using a subscript, overbar, or some other marking to distinguish it from the bulk wave parameters that are derived from the full wave spectra.

**We thank the Reviewer for their comment. We removed the term "mean", as it is the middle value of the chosen sector/bin, as noted by the Reviewer.**

Discussion:
1) On page 14, line7, the authors state, "...the yellow pattern...indicating bed erosion...". It was previously stated that red patterns indicate accretion, and in figure 4, yellow falls on the positive side of the color bar with red. This seems to be contradictory.

**We agree with the Reviewer and changed "yellow" into "light green".**

2) It is unclear what "local" refers to in the discussion of the bar dynamics (page 19, beginning with line 31). Are the local wave parameters reported for a certain point along a chosen profile? If so, which profile and what is the location of the reported local parameters?

**The Reviewer is right: "local" means a specific water depth along the cross-shore section, i.e. $3m$, which is just off the outer bar locations (e.g., see Fig.2). Hence, the waves have been transferred to a $3m$ depth using Goda's method. A new sentence has been included in sect.4.4.**

Presentation:
1) The use of phrases "the former", "the latter", etc. are confusing and cumbersome to comprehension. Please replace these phrases with the intended nouns for a more clear presentation of the work.

**Amended as suggested.**

2) Page 2, line 11: I think the authors intended to use the word "experiments" in place of "experiences".

**Amended as suggested.**

3) Page 6, line 9: The authors do not define DTM.

**Amended as suggested.**

4) Referring to the wave directional sectors by either the abbreviation or the range in degrees would make the text easier to read.

**We apologize to the Reviewer, but we cannot understand which figure or part of the text she/he is referring to, since usually we refer to the most energetic sectors using NNE and ESE.**

5) It would be useful to refer to fig 5 when the green and purple lines are discussed on page 15, line 15.

**Amended as suggested.**

6) Correct the run-on sentence on page 16, line3  page 17, line 2.

**Amended as suggested.**

Figures:

1) It would be useful to show the location of the waverider in the subpanel of Figure 1(a).

**Amended as suggested.**

2) It would be useful to note the profile number of the profiles shown in fig 2.

**The profile number (P13) has been included in the caption.**

3) It would be useful to have the line markers in fig 4 match the markers used of the inner, interm, and outer bars as shown in figure 5.

**Amended as suggested.**

[revised manuscript text omitted]